# Southern Indian Ocean Dipole as a trigger for Central Pacific El Niño since the 2000s

Hyun-Su Jo [1], Yoo-Geun Ham [1]✉, Jong-Seong Kug[2], Tim Li [3,4], Jeong-Hwan Kim [1], Ji-Gwang Kim [1] & Hyerim Kim[5]

Despite decades of effort, predicting the El Niño-Southern Oscillation (ENSO) since the 2000s has become increasingly challenging. This is due to the weaker coupling between the ENSO and well-known precursors in tropical ocean basins, particularly in the Indian Ocean. Here we show that the Southern Indian Ocean Dipole (SIOD), which is characterized by an east-west-oriented sea surface temperature dipole pattern over the southern Indian Ocean, has become a key precursor of Central Pacific El Niño since the 2000s with a 14-month lead. The role of the SIOD in the subsequent year's ENSO is distinctive from the equatorial Indian Ocean Dipole mode in that it prolongs the ENSO period. The westward-shifted ENSO has sustained simultaneous SIOD events for longer periods since the 2000s, which leads to weak but persistent westerly anomalies over the western Pacific. This eventually results in the development of the Central Pacific El Niño in the subsequent year.

The El Niño-Southern Oscillation (ENSO) irregularly fluctuates between El Niño and La Niña conditions in the tropical Pacific with massive socio-economic and environmental impacts worldwide[1–4]. The physical mechanisms underlying the evolution of El Niño and La Niña events and their transitions are reasonably well understood. The occurrence of El Niño events is led by the accumulation of the equatorial warm water volume (WWV) as a result of the slow ocean adjustment process during La Niña events, indicating a lead–lag relationship between the WWV and the ENSO[5,6]. However, this relationship has weakened since the 2000s[7], suggesting that the WWV-related recharge/discharge process driving the transition of the ENSO phase has become less efficient, thus ENSO phase transitions have become less regular compared to the 1979–1999 period. In addition, ENSO behaviors have become increasingly diverse, thus it has become more difficult to predict ENSO events since the 2000s[8–10].

The non-stationary relationship between the ENSO and large-scale oceanic variabilities outside the tropical Pacific can also hinder the successful forecasting of the ENSO[7]. This is evident for the dominant sea surface temperature (SST) variability in the Indian Ocean (IO), which is known to lead the ENSO in the subsequent year[11–14]. Since the early 1990s, the influence of the Indian Ocean Dipole (IOD) and the Indian Ocean Basin-wide mode (IOBM) on the subsequent year's ENSO has significantly weakened[15] (Fig. 1a). On the other hand, SST variability over the southeastern IO has increased considerably since the late 1990s, which can influence tropical Pacific variability through inter-basin coupling mechanisms[16–19]. This may be indicative of the emergence of a new ENSO precursor over the southern IO replacing the previously well-known precursors. In this regard, we propose that the Southern Indian Ocean Dipole (SIOD), which is characterized by an east-west-oriented SST dipole pattern over the southern IO, could be a powerful precursor to Central Pacific (CP) El Niño events occurring after the 2000s.

## Results

The SIOD emerges from the southern IO during the October-November-December (OND) season during the 1998–2019 period as

[1]Department of Oceanography, Chonnam National University, Gwangju, South Korea. [2]Division of Environmental Science and Engineering, Pohang University of Science and Technology, Pohang, South Korea. [3]International Pacific Research Center and Department of Atmospheric Sciences, School of Ocean and Earth Science and Technology, University of Hawaii at Manoa, Honolulu, Hawaii, USA. [4]Key Laboratory of Meteorological Disaster, Ministry of Education (KLME)/Joint International Research Laboratory of Climate and Environmental Change (ILCEC)/Collaborative Innovation Center on Forecast and Evaluation of Meteorological Disasters (CIC-FEMD), Nanjing University of Information Science and Technology, Nanjing, China. [5]Department of Marine Sciences and Convergent Technology, Hanyang University, ERICA, Ansan, South Korea. ✉e-mail: ygham@chonnam.ac.kr

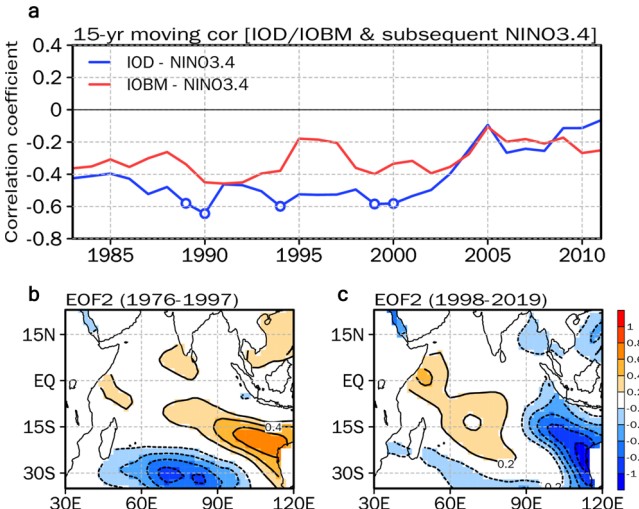

**Fig. 1 | The emergence of a new climate mode over the Southern Indian Ocean after the 2000s. a** 15-yr moving correlation coefficients between the Indian Ocean Dipole (IOD) index during boreal fall (SON) season, the Indian Ocean basin mode (IOBM) index during boreal winter (DJF) season, and the subsequent year's Niño3.4 index during the DJF season (blue line: IOD vs. Niño3.4; red line: IOBM vs. Niño3.4) from 1976–2019. The x-axis indicates the middle year in the 15-yr moving window (e.g., 1999 indicates the correlation coefficient from 1992–2006). The dots denote correlation coefficients that are statistically significant at the 95% confidence level. **b** Second empirical orthogonal function (EOF) of sea surface temperature (SST) (°C) anomalies over the Indian Ocean during boreal late fall (OND) season for 1976–1997. **c** As in **b** but for 1998–2019. The units for the EOF are non-dimensional.

a 2nd empirical orthogonal function (EOF) of the SST anomalies over the IO (Fig. 1c), differing from the Indian Ocean Subtropical Dipole (IOSD)[20,21] which appears as a 2nd EOF during the 1976–1997 period (Fig. 1b). The SIOD is also captured as a dominant mode in self-organizing maps (SOM)[22], a nonlinear statistical method, indicating that the SIOD is a physical mode over the southern IO (Fig. S1c, e). Based on SIOD-related eigenvectors (i.e., the 2nd EOF for 1998–2019), we define the SIOD index by taking the SST anomaly differences between the southcentral IO [65°–85° E, 25°–10° S] and southeastern IO [90°–120° E, 30°–5° S] (Fig. 2c). Note that the SIOD is independent of the IOSD and the IOD in terms of the temporal lead−lag correlation (Fig. S2) and the season of peak intensity (Fig. S3). In addition, even though there is a similarity in the spatial distribution between the positive SIOD and Ningaloo Niña events[23,24] (Fig. S4), the physical mechanisms of the two are distinct (Figs. S5 and S6).

To demonstrate the newly emerging role of the SIOD on the subsequent year's ENSO since the 2000s, lagged regression is conducted on the SIOD index (Fig. 2c, d), and this is compared to regression on the IOD index, which is defined as the SST anomaly differences between the western [50°–70° E, 10° S–10° N] and eastern IO [90°–110° E, 10–0° S] (Fig. 2a). Note that all of the indices are detrended before the analysis by removing the linear trend within each analyzed period (i.e., 1976–2019, 1976–1997, and 1998–2019), and the regression coefficients reflect a one standard deviation change in the indices. The SST anomalies regressed on the IOD index from 1998–2019 exhibit positive and negative SSTs over the equatorial western-eastern IO during the September-October-November (SON) season (Fig. 2a) where the IOD exhibits its peak amplitude (Fig. S3b). Furthermore, positive SST anomalies are observed over the equatorial central-eastern Pacific, indicating that positive IOD events tend to co-occur with El Niño events[25]. After 15 months, a neutral state is present over the equatorial central-eastern Pacific[14,26] (Fig. 2b), confirming that the impact of the IOD on the subsequent year's ENSO has weakened since the 2000s[15] (Fig. 1a).

In terms of the SIOD during the OND season from 1998–2019, the spatial distribution of SST anomalies is similar to the positive IOD with a southward shift (Fig. 2c). As with the IOD, positive SIOD events tend to co-occur with positive SST anomalies over the equatorial western-central Pacific, except for the stronger poleward-extended SST anomalies over the equatorial Pacific, which are possibly due to rapid positive feedback between the CP El Niño and the Pacific Meridional Mode[27]. This simultaneous relationship is also robust for individual cases, though a few exceptional cases of the coexistence of a positive SIOD and La Niña events (i.e., 2008, 2017) are also observed.

Intriguingly, the SIOD-induced SST anomalies over the equatorial Pacific in the winter of the subsequent year completely differ from those induced by the IOD. After 14 months, positive SIOD events induce El Niño events, which is significant at a 95% confidence level (Fig. 2d). In particular, the zonal action center for the positive SST anomalies is located over the equatorial central Pacific, indicating that a positive SIOD can induce a CP-type El Niño event in the subsequent year. The lag-relationship between the SIOD and the ENSO is characterized by completely different decadal modulation from that between the IOD and ENSO. The negative lag-relationship between the IOD and Niño3.4 (SSTs averaged over 170°–120° W, 5° S–5° N) in the subsequent winter is significant for 1976–1997 (the blue bar in Fig. 2e), while the positive relationship between the SIOD and Niño3.4 is significant for 1998–2019 (the red bar in Fig. 2e). This indicates the emergence of the SIOD as a precursor of the ENSO after the 2000s, since when the impact of the IOD has significantly weakened.

The role of the SIOD in the development of the subsequent year's ENSO is more pronounced than that of the southeastern IO (SEIO) mode[28,29] or Ningaloo Niño/Niña[23], which are characterized by cold/warm SST anomalies during the boreal winter or early spring season in the southeast IO off Australia. The SEIO acts as a precursor of the ENSO for 9 months[28], which is shorter by 5 months than that of the SIOD. Therefore, the correlation coefficient between the SEIO during the OND season and Niño3.4 during the December-January-February (DJF) season in the subsequent year is only −0.25 (p−value=0.24) for 1998–2019. Similarly, the influence of Ningaloo Niño/Niña during the OND season on the subsequent year's ENSO is negligible at the 95% confidence level (Fig. S7). The stronger impact of the SIOD on ENSO events in the subsequent year compared to that of Ningaloo Niño/Niña is because both the western and eastern loading of the SIOD has an additive effect on the ENSO (Fig. S8).

The lead−lag correlation between the SIOD during the OND season and the Niño4 (SSTs averaged over 160° E-150° W, 5° S–5° N) shows two positive peaks at a lag of zero (i.e., OND(0), where (0) denotes the simultaneous year for the SIOD peak phase) and 13 months (i.e., ND(+1)J(+2), where (+1), and (+2) denote one, and two years after the SIOD peak year, respectively) (the red line in Fig. 2f), while the correlation coefficients between the SIOD and Niño3 (SSTs averaged over 150°–90° W, 5° S–5° N) are not statistically significant at any lag months (the blue line in Fig. 2f). This clearly demonstrates that the SIOD has significantly contributed to the multi-year persistence of SST anomalies over the equatorial central Pacific for 1998–2019. The SIOD can thus be considered a precursor for the prolonging of the ENSO signal, while the precursors outside the tropical Pacific contribute to accelerating the ENSO phase transition[14,30,31]. It should be noted that the auto-correlation of Niño3 and Niño4 gradually decays during the following year and eventually transits into a neutral state in the winter of the subsequent year (the gray and black lines in Fig. 2f), indicating that the Niño3 and Niño4 SSTs themselves cannot easily persist for two consecutive years.

To understand the mechanisms responsible for the positive SIOD facilitating the development of El Niño events in the subsequent year, we conducted simultaneous regressions on the SIOD index (Fig. 3a, b). During the OND season, the cold SST anomalies over the southeastern IO reduce the precipitation over the maritime continent (Fig. 3a). This

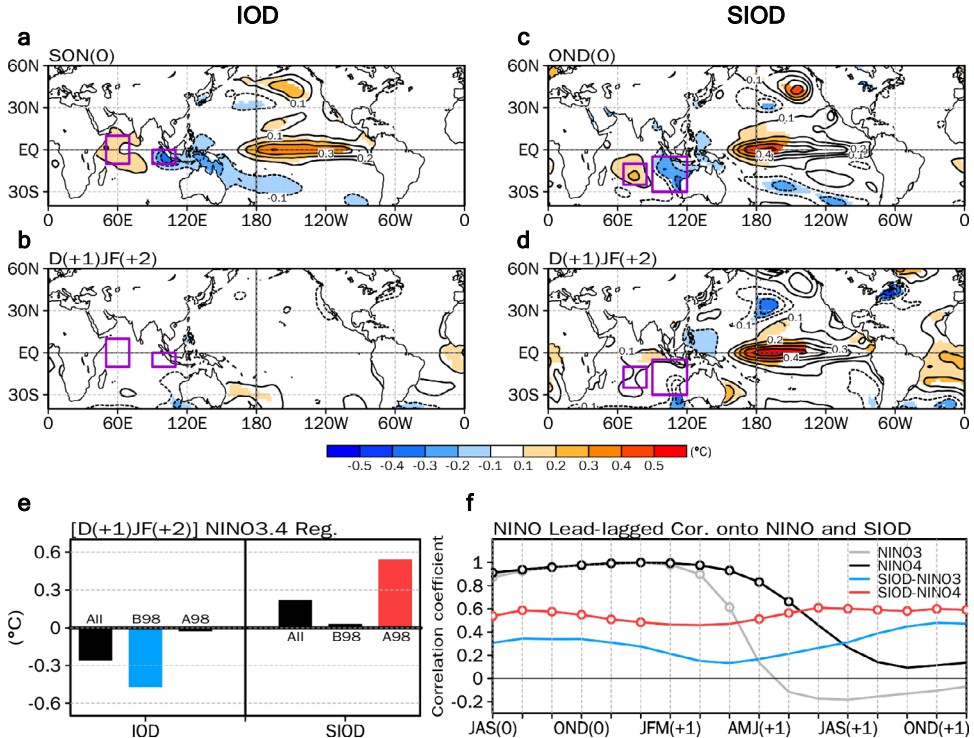

**Fig. 2 | Southern Indian Ocean Dipole as a precursor of the ENSO after the 2000s.** Lagged regression maps of (**a**) sea surface temperature (SST) (°C) anomalies during boreal fall (SON) season and (**b**) subsequent year's SST (°C) anomalies during boreal winter (DJF) season with respect to the normalized SON Indian Ocean Dipole (IOD) index from 1998–2019. (**c**) SST (°C) anomalies during boreal late fall (OND) season and (**d**) subsequent year's SST (°C) anomalies during the DJF season with respect to the normalized OND Southern Indian Ocean Dipole (SIOD) index from 1998–2019. Shading in **a**–**d** denotes the region where the statistical significance is above the 95% confidence level. **e** Regressions between the IOD, SIOD,

and Niño3.4 SST index during the subsequent year's DJF season for 1976–2019 (left), 1976–1997 (middle), and 1998–2019 (right), respectively. The blue and red bars in **e** indicate regression coefficients that are statistically significant at the 95% confidence level for negative and positive coefficients, respectively. **f**, Lead-lagged regression between the Niño3 SST and SIOD index (blue), the Niño4 SST and SIOD index (red), and auto lead-lagged regression of the Niño3 SST (grey) and Niño4 SST indices (black) from OND(0) to the subsequent year's boreal early winter (ND(+1) J(+2)) season for 1998–2019. The open circles in **f** indicate correlation coefficients that are statistically significant at the 95% confidence level.

negative precipitation with a descending motion generates low-level divergent flow over the maritime continent, inducing westerly anomalies over the equatorial western Pacific (Fig. 3b). These westerly anomalies can be also understood as a stationary Kelvin wave response to the negative diabatic heating anomalies led by the reduced precipitation amount over the maritime continent[32].

Because it can be argued that the SIOD-associated SST warming over the equatorial central Pacific is responsible for the equatorial westerly over the western Pacific (Fig. 2c), we conducted a partial regression on the equatorial (averaged over 5° S–5° N) 925 hPa zonal wind (U925) anomaly over the Pacific during the following March-April-May (MAM(+1)) season with respect to the OND(0) SIOD after excluding the impact of D(0)JF(+1) Niño4 (the red line in Fig. 3c). As a comparison, the U925 anomaly during the MAM(+1) season regressed on the OND(0) SIOD is also presented (the blue line in Fig. 3c). The partial regression produces similar amplitudes to those from the conventional regression over the equatorial western Pacific, suggesting that the positive SIOD over the southern IO leads to westerly anomalies over the equatorial western Pacific, while the influence of SST warming over the equatorial central Pacific is minimal. Furthermore, the westerly anomalies regressed on the OND(0) SIOD after removing the impact of D(0)JF(+1) Niño4 gradually intensify over time (the blue line in Fig. 3d); however, when they are regressed on the D(0) JF(+1) Niño4 decays after removing the impact of the OND(0) SIOD (the red line in Fig. 3d). This shows that the westerly anomalies over the equatorial western Pacific during the positive SIOD events are mostly led by the SIOD SSTs, rather than the equatorial central Pacific SST anomalies.

The westerly anomalies increase the ocean heat content (OHC) anomalies over the western-central Pacific during the following year's boreal summer by raising the sea surface height and deepening the thermocline depth along the equator (Fig. S9). As the increased OHC anomalies are robust over the equatorial central Pacific, the warm SSTs and westerly anomalies, which are developed further by positive Bjerknes feedback[33], are also robust over the western-central tropical Pacific. Consequently, the maximum center for the warm SST anomalies is present over the equatorial western-central Pacific in the subsequent year's winter (Fig. 2d).

The El Niño events led by the SIOD (2002, 2003, 2004, 2009, and 2018) reflect a weak contribution from thermocline feedback (i.e., vertical advection led by mean upwelling and the anomalous temperature gradient) and a significantly stronger contribution from zonal advective feedback (i.e., zonal advection of the mean SST by anomalous zonal currents) over the equatorial western-central Pacific when compared with El Niño events without SIOD forcing (2006, 2014, and 2015) (Table S1), as in most CP-type El Niño events[10,34]. The weaker amplitude of SIOD-related thermocline feedback might be caused by the occurrence of simultaneous El Niño during the positive SIOD, leading to a discharge of equatorial heat via poleward geostrophic currents[35]. As the subsurface temperature anomalies over the equatorial Pacific have negative values due to heat discharge, neglectable positive vertical temperature advection is induced. Additionally, the SIOD-related westerly anomalies over the equatorial western Pacific are persistent but relatively weak in terms of their amplitude compared to that of conventional El Niño events (Fig. S10). Therefore, the oceanic Kelvin wave response does not propagate as far east as it does

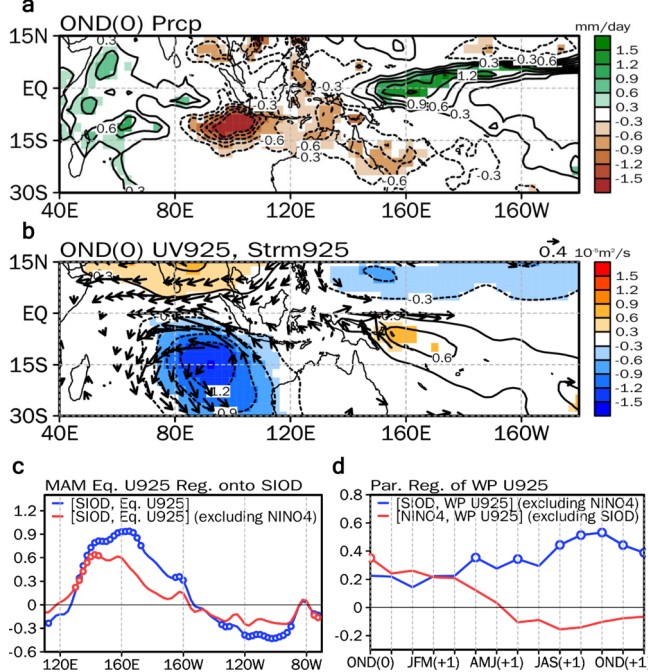

**Fig. 3 | Atmospheric variance associated with the Southern Indian Ocean Dipole.** Regression maps of (**a**) precipitation (mm day⁻¹) and (**b**) 925hPa wind (vector, m s⁻¹) and streamfunction (contours at intervals of $0.3 \cdot 10^{-5}$ m² s⁻¹) anomalies during boreal late fall (OND) season with respect to the normalized OND Southern Indian Ocean Dipole (SIOD) index for 1998–2019. The shading and vector in **a**, and **b** denote the region where the statistical significance is above the 95% confidence level. **c** Regression of 925 hPa zonal wind anomalies averaged over 5° S–5° N over the Pacific during the following boreal spring (MAM) season with respect to the normalized OND SIOD index for 1998–2019 (blue line). The red line is the same but for the partial regression after excluding the impact of the boreal winter (DJF) Niño4 index. **d** Partial lag-regression of 925 hPa zonal wind anomalies averaged over 120°–150° E, 5° S– 5° N over the equatorial western Pacific from the OND season to the subsequent year's boreal early winter (NDJ) season with respect to the normalized OND SIOD index after excluding the impact of the DJF Niño4 index for 1998–2019 (blue line). The red line is the same but for the partial lag-regression with respect to the normalized DJF Niño4 index after excluding the impact of the OND SIOD index for 1998–2019. The open circles in **c**, and **d** indicate correlation coefficients that are statistically significant at the 95% confidence level.

during conventional El Niño (Fig. S11). In turn, positive SIOD-induced El Niño events tend to be of the CP type.

Although this observational analysis demonstrates the role of the SIOD in the subsequent year's CP-type El Niño event, it has limitations due to the relatively short period of analysis and the combination of various climate variabilities. Therefore, the impact of the SIOD is further substantiated using idealized experiments with an atmosphere-ocean coupled general circulation model (CGCM) (see Table S2 for the experimental design). The differences between the CGCM experiments with and without the SIOD-related SST anomalies in the southern IO (Fig. 4a) confirm that the positive SIOD-related SST leads westerly anomalies over the equatorial western Pacific during the OND season (Fig. 4b), inducing an El Niño event in the subsequent year's winter (Fig. 4c). This is also further supported by AGCM simulations with identical SIOD-related SST forcing (Fig. S12) as also reported by ref. 16.

Taken together, the results above suggest that the SIOD is a powerful precursor that can significantly improve ENSO forecasting skills. We construct a regression model using the SIOD index for the OND season to predict the Niño index, and its forecasting skill is compared with a bilinear regression model whose predictors are the Niño and OHC indexes during the winter from 1998–2019 (Fig. 4d, e). With an approximate six-month lag, the hindcast correlation skills of

the Niño3, Niño3.4, and Niño4 indexes with the precursor of the SIOD index are systematically superior to those with a combination of the OHC and Niño indices. In particular, while the correlation skills of Niño3.4 or Niño4 with the SIOD are high (> 0.4) with a lead time of up to 13 months (Fig. 4d), the correlation skills of the Niño indices with the OHC and Niño indexes are nearly zero with a year-lead forecast (Fig. 4e). The role of the IOD in predicting subsequent El Niño events for 1998–2019 is also minimal (Fig. S13). Because the forecast skill of Niño4 is systematically superior to that of Niño3.4, the SIOD becomes a key precursor for CP-type El Niño events after the 2000s.

The SIOD-related SST anomalies persist for longer (roughly five seasons longer) during 1998–2019 than during 1976–1997 (Fig. 5a). Consequently, the SIOD-induced westerly anomalies over the equatorial western Pacific since the 2000s can be sustained until summer the following year and thus consistently affect the subsequent ENSO evolution (Fig. 3c, d). The decadal difference in the spatial distribution of the ENSO is possibly a cause of the multi-season persistence of the SIOD-related SST anomalies since the 2000s by shifting its warming center to the west[9,36]. Associated with the westward shift in the SST warming center, the overall ENSO-related atmospheric responses also shift westward, reaching the IO basin. To mimic the post-2000 ENSO, whose zonal action center shifts to the west compared to before the 2000s (Fig. S14), we conducted a composite analysis of the two types of ENSO events (i.e., Eastern Pacific (EP) and Central Pacific (CP) ENSO) following definitions from a previous study[37] (see Methods).

The maximum center of the composited SST anomalies for the CP ENSO significantly shifts to the west compared to that of the EP ENSO (Fig. S15a, b). The anticyclonic circulation is observed over the southern IO in both composite maps, but the southerly anomalies over the eastern edge of this anticyclonic circulation are relatively weak in the EP ENSO as it is located in the continental area over western Australia (Fig. S15a). On the other hand, stronger southerly winds off the west coast of Australia are evident for the CP ENSO (Fig. S15b). As a result, the SIOD-related cold SST anomaly over the eastern part of the southern IO is more robust in the CP ENSO than in the EP ENSO due to the higher wind speed and resultant latent heat flux[24,38] (Fig. 5b, d).

These results are supported by two other idealized CGCM experiments, the Exp_C_EPEN and Exp_C_CPEN experiments, which are forced by SST anomalies over the tropical Pacific associated with the EP ENSO and CP ENSO, respectively (Table S2 and Fig. S15c, d). The difference between the Exp_C_EPEN and Exp_C_CPEN experiments indicates that the westward-shifted El Niño zonal warming center contributes to sustaining SIOD-related cold SST anomalies over the southeastern IO (Fig. 5c, e). In turn, these long-lasting SIOD-related SST anomalies have strongly influenced the evolution of the ENSO since the 2000s (Fig. 3), indicating that the inter-basin coupling between the IO and Pacific sector contributes to sustaining both the SIOD and the ENSO for multiple seasons after reaching its peak phase[18].

## Discussion

This study highlights the impact of the SIOD on the subsequent year's ENSO through observational analysis and a series of CGCM experiments. Although the spatial distribution of the SIOD resembles that of the IOD with a latitudinal shift, the SIOD has the opposite impact on the subsequent year's ENSO. The IOD helps accelerate the phase transition of the ENSO within a year, whereas the SIOD promotes the persistence of the ENSO for two consecutive years. Therefore, the SIOD possibly contributes to the long-lived SST anomalies over the central Pacific[10], which are responsible for the Pacific decadal variability[39,40]. Additionally, the impact of the SIOD on the ENSO has become prominent since the 2000s, while the impact of the IOD has abruptly reduced in recent decades[15]. This change is accompanied by the westward shift in the ENSO-related SST variability, which enhances the simultaneous ENSO–SIOD coupling and the resulting long-lasting SIOD events since the 2000s (Fig. 5a).

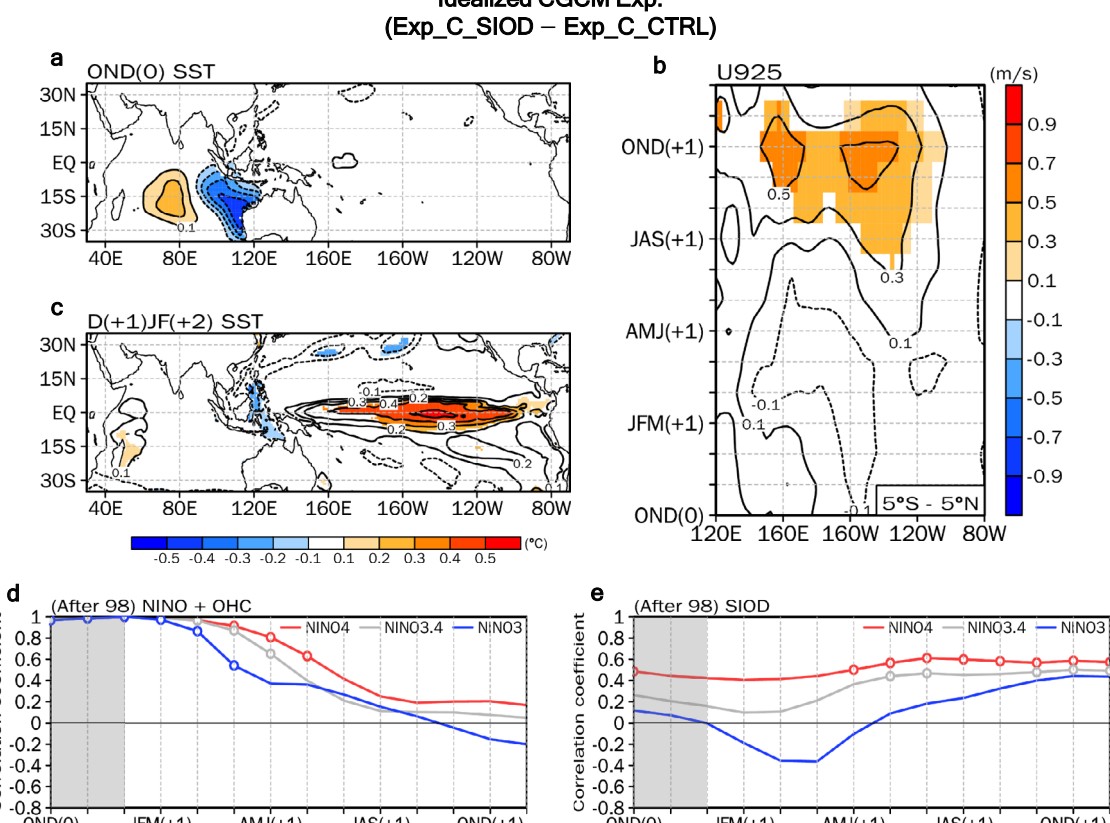

**Fig. 4 | Idealized coupled model experiments and correlation skill of the Southern Indian Ocean Dipole index as an ENSO precursor.** Changes in the (**a**) precipitation (shading, mm day$^{-1}$) and 925 hPa wind (vector, m s$^{-1}$), and (**b**) sea surface temperature (SST) (°C) anomalies in the Southern Indian Ocean Dipole (SIOD) experiment (Exp_C_SIOD) compared to the control experiment (Exp_C_CTRL) during boreal late fall (OND) season. **c**, **d**, As in **a**, **b**, but for during the subsequent year's winter. The shading in **a**–**c** denotes the region where the statistical significance is above the 90% confidence level. **e** Hindcast correlation skills of

Niño4 SST (red), Niño3.4 SST (grey), and Niño3 SST (blue) indices from the OND season to the subsequent year's boreal early winter (NDJ) season using a combination of the boreal winter (DJF) Niño and the DJF ocean heat content (OHC) indices for 1998–2019. **f** As in **e**, but using the OND SIOD index. The grey box indicates the developing seasons of the SIOD. The open circles in **d**, and **e** indicate correlation coefficients that are statistically significant at the 95% confidence level. A leave-one-year-out cross-validation method is employed.

The SIOD may provide an additional avenue for the prediction of the ENSO, in particular the CP-type ENSO, which is more challenging than predicting the EP-type El Niño[41]. As the SIOD modulates both the spatial and temporal characteristics of the ENSO, advances in research towards a better understanding of the role of the SIOD may strengthen its potential as a unique and reliable predictor for CP-type El Niño events in recent decades. Therefore, in addition to the countries adjacent to the equatorial Pacific, the main findings of this study have the potential to enhance the forecasting ability for global climate variabilities led by various types of El Niño-induced atmospheric teleconnections[9,36].

## Methods
### Observations
The monthly mean SST data from the Extended Reconstructed Sea Surface Temperature version 5 (ERSST v5)[42] were used. The monthly mean precipitation was obtained from the Global Precipitation Climatology Project (GPCP)[43]. The monthly mean 925hPa winds and streamfunction were obtained from the National Centers for Environmental Prediction-National Center for Atmospheric Research (NCEP–NCAR) reanalysis 1[44]. The monthly OHC for the upper 300 m was derived from the European Centre for Medium-Range Weather Forecasts Ocean Reanalysis System 5 (ORAS5)[45]. The analysis period spans from 1976 to 2019 because the period, amplitude, spatial structure, and temporal evolution of El Niño and its relationship with the Indian Ocean SST notably changed after the mid-1970s[28,46].

### Statistical analysis and significance testing
Hindcasts of the three-month averages of the Niño indices in Fig. 4 were employed using predictors (a combination of Niño and OHC indices and the SIOD index) from OND(0) to ND(+1)J(+2) for 1998–2019. The widely used one-year-leave-out cross-validation method was applied for the hindcasts by excluding the observed value for the target year. Two-tailed Student's $t$-tests were used to determine statistical significance[47] for regression and correlation analysis. Furthermore, statistical significance testing for the composite analysis was conducted according to the bootstrap method[48].

### Coupled Global Climate Model (CGCM)
To investigate the impact of the SIOD on the subsequent year's ENSO during the winter, the Community Earth System Model, version 1 (CESM1)[49] was used for the idealized CGCM experiments. The model resolution was a 1.9° × 2.5° grid with the standard 30 vertical levels for the atmosphere and an approximate 1° grid for the ocean.

We first performed two partial nudging experiments. The nudging scheme is one of the simplest data assimilation techniques and can be described as follows:

$$\frac{dSST}{dt} = A + \frac{(SST_{obs} - SST)}{\tau} \tag{1}$$

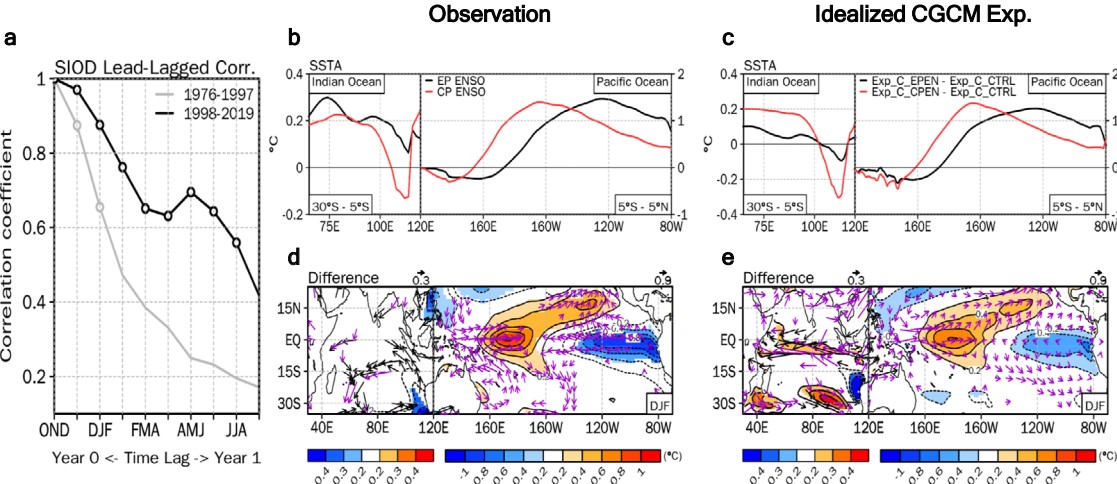

**Fig. 5 | Impact of the westward-shifted ENSO on the Southern Indian Ocean Dipole (SIOD) variability after the 2000s. a** Auto-lagged correlations of the boreal late fall (OND) SIOD index from the OND season to the subsequent year's boreal late summer (JAS) season for 1976–1997 (gray line) and 1998–2019 (black line), respectively. The open circles in **a** indicate correlation coefficients that are statistically significant at the 95% confidence level. **b** Latitudinally averaged observed sea surface temperature (SST) (°C) anomaly composite for the Eastern Pacific (EP) (black line) and Central Pacific (CP) ENSO (red line) events during the OND season for 1998–2019 over 30°–5° S in the southern Indian Ocean (left panel),

and over 5° S–5° N in the equatorial Pacific (right panel). **c** As in **b**, but for changes in the latitudinally averaged SST (°C) anomaly in Exp_C_EPEN (black line) and Exp_C_CPEN (red line) compared to the control experiment (Exp_C_CTRL). **d** Observed difference in the composited SST (°C), and 925 hPa wind (vector, m·s⁻¹) anomalies for EP and CP ENSO events during the OND season from 1998–2019. **e** As in **d**, but for the difference in Exp_C_EPEN from that in Exp_C_CPEN. The shading and purple vector in **d**, and **e** denote the region where the statistical significance is above the 90% confidence level.

where $SST$, and $A$ represent the $SST$ in the CGCM, and the conventional dynamical and physical terms to lead SST tendency, respectively. The second term on the right-hand side is the nudging term. By adding the nudging term, the simulated SST always moves toward the observed SST. The nudging time scale ($\tau$) was prescribed as 1 day so that the simulated SST state in the southern IO was strongly nudged toward the given observed SST state.

In the first experiment, the positive SIOD-related SST anomalies were nudged in the southern IO (60°–120° E, 30°–5° S) with the seasonally varying climatological SST obtained from ERSST v5 for 1998–2019, whereas the other oceans were run freely (Exp_C_SIOD). The second experiment was used as a control, which was identical to the former experiment except that the observed climatological SST anomalies were nudged in the southern IO (Exp_C_CTRL). To obtain the positive SIOD-related SST anomaly pattern, we first collected the years in which the OND SIOD index was larger or small than one standard deviation for the composite analysis. For these years, we calculated the difference in the composite between the positive and negative SIOD events, and then the values were divided by 2 to obtain the final SIOD-related SST anomaly pattern. Finally, the composited SST anomalies were added to the climatological SST. A total of 30 ensemble members with different initial conditions were employed for both experiments. Each experiment was run for 17 months from October (0) to February (+2).

Second, we performed two additional partial nudging experiments to investigate the impact of the westward-shifted ENSO on SIOD variability after the 2000s. These were identical to the SIOD experiments except that the EP and CP El Niño-related SST anomalies were nudged in the tropical Pacific (120° E–80° W, 20° S–20° N) (Exp_C_EPEN and Exp_C_CPEN, respectively). We followed the definitions of the EP and CP ENSOs from a previous study[37]. We first collected the years for which the $N_{CT}$ or $N_{WP}$ index during the winter was greater (less) than 0.5 °C (−0.5 °C). Note that the $N_{CT}$ and $N_{WP}$ indices were calculated using a mathematic rotation of Niño3 and Niño4 indices. Of these years, an EP El Niño (La Niña) year was defined as when the $N_{CT}$ index was greater (less) than the $N_{WP}$ index. We then calculated the

difference in the composite between the EP El Niño and La Niña cases and the values were divided by 2 to obtain a final EP ENSO pattern. The CP ENSO pattern was also obtained using the same method for the EP ENSO except that a CP El Niño (La Niña) year was defined as when the $N_{WP}$ index was greater (less) than the $N_{CT}$ index. A total of 20 ensemble members with different initial conditions were employed for both experiments. Each experiment was run for five months from October (0) to February (+1).

**Atmospheric Global Climate Model (AGCM)**

The Community Atmosphere Model version 5 (CAM5)[50] was used for the idealized AGCM experiments to investigate the impact of the SIOD on the wind anomalies over the equatorial western Pacific. The model had a horizonal resolution of 1.9° × 2.5° with 30 vertical levels. Two AGCM experiments were conducted: one was a control (Exp_A_CTRL) and the other was a sensitivity experiment (Exp_A_-SIOD). Exp_A_CTRL was forced by the monthly climatological SST (1976–2019) over the globe. Exp_A_SIOD was identical to Exp_A_CTRL except that the positive SIOD-related SST anomalies were prescribed in the southern IO (30°–130° E, 30°–0° S). To obtain the positive SIOD-related SST anomaly pattern, the observed SST anomalies were linearly regressed on the OND SIOD index from 1998–2019; subsequently, these regressed SST anomalies were added to the climatological SST field (Fig. S12a). A total of 35 ensemble members with different initial conditions were employed for both experiments. Each experiment was run for one year from October (0) to September (+1).

## Data availability

The ERSST dataset is available at https://psl.noaa.gov/data/gridded/data.noaa.ersst.v5.html. The GPCP dataset is available at https://psl.noaa.gov/data/gridded/data.gpcp.html. The NCEP/NCAR monthly reanalysis is available at http://www.esrl.noaa.gov/psd/data/gridded/data.ncep.reanalysis.html. The ORAS5 dataset is available at https://cds.climate.copernicus.eu/cdsapp#!/dataset/reanalysis-oras5?tab=form.

# Article

## Code availability

The data in this study were analyzed with NCAR Command Language (NCL; http://www.ncl.ucar.edu/). The code of the CESM1.2 model used in this study is available at http://www.cesm.ucar.edu/models/cesm1.2. All relevant codes used here are available, upon request, from the corresponding author.

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

## Acknowledgements

We thank the anonymous reviewers for their valuable comments. This research was supported by the Basic Science Research Program through the National Research Foundation of Korea (NRF) funded by the Ministry of Education (NRF-2021R1I1A1A01053579) Y.-G. Ham was supported by the National Research Foundation of Korea (NRF)(NRF-2020R1A2C21010025). T.L. was supported by NSF AGS-2006553.

## Author contributions

H.-S.J. and Y.-G.H. designed the research, conducted the analysis, and wrote the majority of the manuscript. J.-S.K. and T.L. contributed significantly to the interpretation of the analysis and model results. J.-H.K., J.-G.K., and H.K. performed the experiments and analyzed the data. All of the authors discussed the results and reviewed the manuscript.

## Competing interests

The authors declare no competing interests.
