## [Peer Review File · Nature Communications]

Southern Indian Ocean Dipole as a trigger for Central Pacific El Niño since the 2000sREVIEWER COMMENTS

Reviewer #1 (Remarks to the Author):

Main comments

This manuscript is about the predictability of ENSO and the possible changes of the key-predictors of El Niño and La Niña events in the two last decades.

As noted already in the literature, they first illustrate that the traditional ENSO precursors, the Warm Water Volume (WWV) in the Pacific and the Indian Ocean (IO) tropical modes (e.g. the IOD and IOB) modes) lose their significance since the 2000s as their relationships with the low-level zonal wind anomalies over the western Pacific have weakened significantly recently. The authors suggest that this evolution is related to the emergence of a new mode of variability in the south subtropical IO, the Southern Indian Ocean ¹¹_{SEP}Dipole (hereafter SIOD), which is distinct from the tropical IO modes and the Indian Ocean Subtropical Dipole (IOSD hereafter) located more to the South (Behera and Yamagata 2001), which have already been largely studied in the literature and are known to be both partly forced by ENSO and possible ENSO precursors.

Based on statistical analysis and different climate model experiments (with both coupled and atmospheric forced climate models), they suggest that the SIOD is now an active player in ENSO evolution and a key precursor of ENSO.

However, the results and arguments presented in the manuscript fall short to be convincing to me for several reasons:

- First, the definition of a new climate (e.g., SST here) index requires the demonstration that this time series represents a coherent physical mode of variability or at least a mode, which represents a substantial part of the SST variability in the (Indian) domain.

Obviously, the SST variability in the tropical IO is dominated by the IOD (and the IOB) and the IOSD in the subtropics and they are both intimately related, even without ENSO forcing, to sustain biennial variability in the IO (Cretat et al. 2018). The IOD and IOSD peak, respectively, in boreal fall (September and October) and boreal winter (e.g. from December to February), and the authors argue (without any justification) that their SIOD peaks between boreal fall and boreal winter, which corresponds to a season with low SST variability relative to boreal fall or winter in the IO.

Nothing has been presented in the manuscript to justify that the SIOD is not an artificial mode and that its peak season is between the IOD and IOSD peak seasons. Furthermore, the existence of such mode can be hardly justified in the presence of remote ENSO forcing and regional IOD and IOSD forcings without dedicated analysis, and such analysis is clearly missing here.

My interpretation is that its definition has been mainly based to maximize its correlation with ENSO indices since the 2000s. This can be a start, but obviously this must be complemented by convincing arguments that the defined mode is a physical, not artificial, entity and the resulting relationship is not only the product of internal variability.

- Second, the results are largely overemphasized by the authors, as the SIOD seems only related to Niño4 SST and not to Niño3 or Niño34 SST, which are more traditional ENSO indices (see Fig. 3f). This also leads to the alternate hypothesis that SST variability in the central and west Pacific, represented by Niño4 SSTs, drives the SST gradient in the South IO related to the SIOD during recent decades, rather than the reverse (see Fig. 1c). This hypothesis is never explored or refuted in the manuscript. Such possibility can be checked by examining the correlations between the SIOD and SSTs at both positive and negative lags and by examining the persistence of Niño4 and Niño3 SSTs separately in Fig. 1f, which is fully justified as shown in Fig. 3f (rather than using Niño34 SSTs). In other words, the authors must demonstrate that the persistence of Niño4 SSTs by itself does not perform better than their SIOD for predicting Niño4 SSTs itself since the 2000s in observations.

- Third, even if we admit that the SIOD is a coherent mode, the authors were not able to demonstrate clearly that something has changed around 2000 for explaining its emergence as an ENSO precursor. At the end of the manuscript (e.g., lines 193-221 and Figs. 4 and S8), the authors suggest that this is related to the westward shift of ENSO since the 2000s (see Fig. S7), which promotes colder SSTs in the eastern pole of their SIOD index. They justify this by showing differences between regressions with respect to Nino3 and Nino4 indices in observations (Fig. 4d) as well as differences between coupled experiments with Nino3 and Nino4 forcings, respectively (Fig. 4e), which both exhibit cold SST differences off Australia. However, my understanding is that Figs. 4d and e are simply the differences of the panels in each column of Fig. S8. Obviously, from Fig. S8, the differences between the Nino3 and Nino4 forcings over the IO are both very small and insignificant both for observations and model experiments. Moreover, the Nino4 forcing does not really produce cold SST anomalies off Australia (see Fig. S8d), so this argument does not hold. Furthermore, It is not obvious to me how these analyses can explain the decadal variability in the relationships between Nino4 SSTs and the SIOD at the interannual time scale, especially if the standard-deviation of the SIOD is the same before and after the 2000s as noted by the authors.

- Finally, there are a lot of errors, confusion and missing details in the legends of the Figures and in the methods Section, which make a hard job for the reader to understand what is exactly plotted in the figures. To name just a few, it is not clear: when and how, the time series are detrended before the various regression analyses or how these regressions are precisely performed; if the EXPSIOD coupled experiment integrates a double SIOD forcing (as written in Table S2), which can be hardly justified, or a simple SIOD forcing has written in lines 262-280. The description of the coupled model experiments is also lacking important details, are these nudging experiments? What is the length of these experiments?

The legend of Fig. 4 is also particularly misleading as for the experiments what is presented is just differences between experiments and not differences between regressions as stipulated in the legend. Statistical significance of anomalies or differences is missing in many plots, which makes difficult to judge their importance (especially because many of them have small amplitudes), etc.

In summary, in my opinion, the authors defined their dipole index a priori and solely based on a lag correlation analysis of ENSO indices with SSTs before the ENSO peak (not shown in the paper). Otherwise, there is no proper physical justification that this climate index represents a well-defined physical phenomenon. Furthermore, what they call the « peak » season of this index corresponds to a minimum of SST variance in the Indian domain between the IOD and IOSD peak seasons. As the IOD and IOSD are significant precursors of each other, I guess that the SIOD has a strong statistical relationship with both the IOD and SIOD modes (not described here), but at present « the physics » of this index is not defined, at least not in the manuscript in its present form.

In addition, the lead relationship between the SIOD and Nino4 SSTs shown in the manuscript is not sufficient to convince me that the SIOD is not only just the result of the interplay between the remote ENSO forcing and regional forcings (e.g., the IOD and IOSD modes) and that this lead relationship is not just a byproduct of internal variability as the period considered is very short (e.g., only 20 years).

Other comments:

Line 47: remove "that" after indicating, otherwise the sentence is incorrect.

Lines 62-66: It is hard for me to see the logic here and how the "prominent background SST warming" lead to the definition of the new dipole index proposed here as the SIOD.

Lines 67-98 and Fig. 1: What is the logic and physical arguments leading to the definition of the "SIOD" index? This is not clear to me at present. Are the SST time series properly detrended before the correlation analysis? Contrary to what is argued (lines 89-90), the amplitude of the SST anomalies in the Pacific is not stronger with the "SIOD" index, only the correlations seem more significant.

Furthermore, I will be very curious to see, if the Nino4 SST index taken in OND(0) will not lead to a better SST prediction in the tropical Pacific at one year lead than the "SIOD" index. In this respect, it will be more interesting to plot the respective lead-lag correlations with both Nino3 and Nino4 SSTs in Fig.1e and f as it is obvious from Fig. 3e and f, that the "SIOD" index is not a "significant predictor" of Nino3 or Nino34 SSTs, only Nino4 SSTs.

By itself, Figs.1c and d can also be mainly due to the strong persistence of the Nino4 SSTs and their long lasting effects on the IO, rather than an indication of a real precursory signal in the IO! Moreover, the strong persistence of Nino4 SSTs after the 2000s is well documented in the literature, I think.

Lines 99-106: I think that these statements are again misleading as already noted above. First, Fig. 1f and the hypothesis that "SIOD" is a unique precursor of ENSO do not match with the results in Fig.3e-f, which show that both WWV and "SIOD" together fail to produce a useful prediction of both Nino3 and Nino34 SSTs. Taking into account the complex evolution of ENSO, the fact that SST (or any other variable) in some areas has a significant lead statistical relationship with ENSO is not a proof by itself of a physically based precursory relationship. This has been illustrated many times in the literature, for example for the relationship between the Indian Monsoon and ENSO.

Lines 122-141 and Fig. 2: How the ENSO signal has been removed? Only on the "SIOD" index on OND(0)? With what ENSO index: Nino3, Nino34 or Nino4 SSTs? Based on simultaneous covariability or with some lags? These details are critical and are missing here! Furthermore, the tight relationship with Nino4 SSTs during all seasons is clearly seen here, but difficult to see what is forcing what if the plots at negative lags are not shown.

As seen in Fig. 1c, the "SIOD" occurs mainly during ENSO, so the sentence "This implies ^[1]_[SEP] that the atmosphere-ocean coupling processes sustain the cold anomalies over the west coast ^[1]_[SEP] of Australia until the February-March-April (FMA) season when most El Niño events are ^[1]_[SEP] initiated (Fig. 2e). » is difficult to justify here!

Lines 154-165, Supplementary Table S1: Details are missing here, what are the years entering in the composites presented in Table S1? Authors must also compute if the statistics for the different terms in the mixed layer budget are really different for the SIOD+El Nino and El Nino cases, to demonstrate that the processes are really different between the two cases.

Lines 166-176, Table S2 and CGCM experiments (lines 262-280): First, the description of the experiments lacks of details and is confusing. First, are these nudging experiments or the authors have simply inserted the SST forcing in each region in the coupling interface before sending the SST to the atmosphere? What is the length of these experiments, one year with 12 members each? The definition of the EXPSIOD is not the same in lines 262-280 and in Table S2 (where it is suggested that the SIOD forcing has been double), which one is the right one?

Finally, while I fully agree that these experiments suggest that the south Indian Ocean SSTs can potentially forced ENSO, the main effect in the EXPSIOD experiment is to enhance classical El Nino events (as with IOSD forcing during boreal spring), not CP El Nino events (e.g., SIOD forcing), which is radically different from what is seen in the observations, e.g. compare Figs. 3d and f. How the authors interpret or explain these differences? In this respect, it will be illuminating to see the panels corresponding to seasons between OND(0) and JJAA(+1), which are presently not shown in Fig. 3, in order to understand how the classical El Nino are triggered in the EXPSIOD experiment as the westerly wind anomalies over the western Pacific are not significant during OND(0) (Fig. 3a). If the forcing in the EXPSIOD experiment has been double, this must also been clearly stated in the text and not only in Table S2.

Lines 177-186 and Figs 3.e-f: What is the rationale of using WWV during fall here? Please explain. Also what are the respective role of WWV and the "SIOD" during the second period? this is critical to

demonstrate the key role of "SIOD" after the 2000s. Surprisingly, WWV does not lead to any skill full prediction of Nino3 or Nino34 SSTs during the first period as well, probably because WWV has the least persistence during OND(0), I guess. Please try to better illustrate the intrinsic role of SIOD in the ENSO skills!

Lines 188-192: Again, I will be interested to see the same plot as Fig. 4a for the Nino4 SST. Are the time series detrended before computing the auto-correlations?

Lines 282-299, Atmospheric Global Climate Model (AGCM): Line 287, it is written that four experiments have been performed, but only two are described, please correct. Later it is written that the length of the simulations is one year, but finally it is stated that each experiment was integrated for 40 years, difficult to follow! Please correct ...

Figure 2: Correct the legend for the statistical significance indicated by diagonal lines as presently we cannot know what variables are concerned (e.g., SST or heat content for the left panels, and rainfall or 925 hPa stream function for the right panels).

Supplementary Fig. 1a: The text in the panel suggests that the moving correlations are between IOD, IOB and the FMA zonal wind over the western Pacific instead of with Nino34 SST during NDJ as written in the legend, please correct.

Supplementary Figs. 3 and 4: Explain clearly how these panels have been obtained! Are these composite or regression maps? Here the data are detrended, which was not done before, why? Finally, include statistical significance in panels (a) and (b) to help the reader to concentrate on the key-points in each panel.

Supplementary Fig. 8: The color bar is missing.

References:

Behera SK, Yamagata T (2001) Subtropical SST dipole events in the southern Indian Ocean. *Geophys Res Lett* 28:327-330

Cretat, J, P Terray, S Masson and KP Sooraj, 2018: Intrinsic precursors and timescale of the tropical Indian Ocean Dipole: Insights from partially decoupled experiment. *Clim Dyn*, 51:1311-1352

Reviewer #2 (Remarks to the Author):

Review of "New emergence of the southern Indian Ocean Dipole as a trigger for El Niño events since the 2000s" by Hyun-Su Jo et al.

Using the observations and coupled model results, the authors have discussed the inter-basin interaction between the Indian Ocean and the Pacific with focusing on three climate phenomena, i.e., IOD, ENSO and Southern Indian Ocean Dipole (SIOD). Since the interaction among climate modes is becoming a hot topic in the climate research community, I have read through this manuscript with much interest.

They claim that the SIOD is becoming a key precursor of El Niño since the 2000s at a 13-month lead. They also claim that not only has the leading role influencing ENSO changed from IOD to SIOD, but the role itself has changed; IOD accelerates the transition of ENSO, while the SIOD prolongs it for two consecutive years.

Although the conclusion was obtained by rather mediocre analyses, it seems reasonable. My main criticism is that the authors do not properly assess the closely related literature done so far in a somewhat different context. The phenomenon that the authors call SIOD has already been studied

intensively as Ningaloo Niño, and the atmospheric (and even oceanic) inter-basin interaction has also been studied in detail. Also, there is much research on the ENSO diversity (Modoki or Dateline/CP El Niño, whatever authors may call) since Ashok et al. (2007)' detailed work. In this sense, the present work related to those aspects may be classified as a review work. A novel point may be the relationship between SIOD and CP El Niño in the Pacific since the 2000s. Therefore, the storyline needs to be re-written, together with giving credit to the precedent literature in a suitable way rather than following the process of the authors' appreciation of the problem.

It will be necessary to comment on the relationship between the negative SIOD (Ningaloo Niña) and La Nina Modoki. Although it was in 1982/83 before the authors' regime shift, possible comments on the unusual case of coexistence of positive SIOD (Ningaloo Niño) and El Niño seems to inspire readers.

Minor comments:

L. 42-52: ENSO diversity should be touched on at an early stage somewhere here

L. 63-66, L. 107-112: Some readers might think SIOD stands for Subtropical Indian Ocea Dipole as introduced by Behera and Yamagata (2001). See

https://en.wikipedia.org/wiki/Subtropical_Indian_Ocean_Dipole

L. 76-L. 78: Some positive (negative) IODs co-occur with La Niña (El Niño)!

L. 82-91: The statistical correlation has nothing to do with the causality in general. We must be careful in wording.

L. 101: red line \diamond blue line

L. 103: The clear identification of the ENSO diversity was first done by Ashok et al. (2007). Credit should not go to follow-on works

L. 107-112: If the authors claim a new SIOD climate mode as an east-west dipole, this simple description is not enough. As I mentioned already, the anomaly west of the Australia is already known as Ningaloo Niño (Niña) and catalogued as a new climate mode first by Kataoka et al. (2013)

Ref: On the Ningaloo Nino/Nina. 2013, Climate Dynamics. Doi:10.1007/s00382-013-1961-z.

L. 99: Authors should recognize the case in which Ningaloo Niño occurred with El Niño in 1982/83.

L.99-141: Authors should know Tozuka et al. (Climate Dynamics, 2014: doi:10.1007/s00382-013-29044-x). Almost everything here was already discussed there.

L.156: Is Table S1 actually derived by the surface mixed-layer heat budget analysis? What happened to the ocean diabatic processes? Also, why the total percentage exceeds 100%?

L. 173-174: This is exactly what Tozuka et al (2014) did.

L. 198-L. 221: I could not follow the meaning of Nino4 index and Nino3 index in this paragraph. It appears those are different from the conventional usage.

L.203-L.212: I could not follow the description here. Figures look almost the same.

Figure 1: The title is strange from a logical viewpoint. The word $\langle \text{impact} \rangle$ suggests the existence of causality. Figures here just denote correlation. (f), $\langle \text{blue} \rangle \rightarrow \langle \text{red} \rangle$.

Figure 4: $\langle c \rangle$, Same as a > -> $\langle c \rangle$, Same as b>. $\langle e \rangle$, Same as c > -> $\langle c \rangle$, Same as d >.

Reviewer #3 (Remarks to the Author):

The study identifies subtropical Indian Ocean conditions as a skillful predictor for ENSO since the 2000s. Using observations, CGCM and AGCM experiments, the utility of the SIOD precursor is evaluated and potential mechanisms for the skill are detailed. Given ongoing challenges with skillful ENSO predictions in recent years, the study tackles a timely and relevant problem in ENSO forecasting. The paper is well-structured and clearly written overall. However, there are inconsistencies in the analyses in different parts of the study that need to be resolved and further clarification required on several aspects (see below comments).

Main comments

- Inconsistencies exist with regard to seasons, analysis periods etc. in different analyses in the main part of the manuscript and supplement. Or at least, deliberate choices for different seasons (e.g., SON

in Fig 1a vs OND in Fig 1c, but then NDJ for both b and d; or different analysis periods for EN and EN+SIOD years in Table S1) must be better motivated and justified in the text.

- It is unclear why the analysis period 1976-2019 was chosen; it does not coincide with limited data availability of the used datasets, as most go back further (though, the GODAE-based WWV is only available post-1980). Suggest using a longer analysis period or provide further justification for this start period. In fact, it appears that the analyses periods coincide with the mid-1970s climate shift in the Indo-Pacific and/or IPO phasing. Yet, no mention is made anywhere as to the implications these factors might have for the results obtained here, or for either ENSO predictability, ENSO diversity, interbasin connectivity etc. This is a missed opportunity to embed the results obtained in this study into a broader discussion of multi-decadal variability in the Indo-Pacific and implications for ENSO characteristics and predictive skill.

Specific comments

- The writing throughout the manuscript would benefit from editing by a native English speaker. Some specific corrections are indicated below, but that is not an exhaustive list.

- L33-34: For those unfamiliar with the IOD, the description of SIOD in the abstract is unhelpful. Suggest describing the SIOD pattern/anomalies in its own right without invoking another climate mode whose pattern then needs to be shifted latitudinally by a certain amount to achieve the overall pattern. This is even more apparent given that the next sentence refers to other factors (sign, timing etc) that differ between IOD and SIOD as precursors for ENSO.

- L36: It is unclear whether the "it" at the end of the sentence refers to ENSO or its transition. Please reword.

- L47: A verb is missing in this sentence; "... lead-lag relationship exists between...?"

- L50: "less periodical" – this is vague and unclear. Does this refer to El Niño-La Niña transitions or periodicity between subsequent El Niño events or persistence of events? Please clarify.

- L67-69: Sentence is grammatically incorrect.

- L71: The boxes identified for the SIOD here differ significantly from earlier SIOD definitions. While this is acknowledged later in the manuscript, it would be helpful to provide some justification for the chosen regions here.

- L74: Unclear why this analysis period 1976-2019 was chosen; see main comment above.

- L85: Reword to "... SIOD co-occurred..."

- L89-91: Even more strikingly, the significant SST anomalies are not just located in the CP region (rather than EP), but at the poleward edges rather than along the equator. Further discussion of these poleward displaced features and their implications is required. What does the zonally broader SST anomalies imply for mechanisms/dynamics/feedbacks?

- L113: This is the first mention of the SEIO mode; please provide further explanation as to its pattern, mechanism and importance in the context here.

- L118: Please cite p-values for the correlations

- L250: WWV data is not available for this time period.

Figures & Supplementary material

- Fig. 2: Areas indicating significant OHC are pretty small and disparate. Would they pass a field significance test? Same applies to precipitation.

- Fig. 3: What is the focus on JJA for the CGCM results? In the observations, NDJ (Fig. 1) or DJF/FMA (Fig. 2) was used.

- Fig. 4a-c: The legend text in grey is nearly invisible in a printed version; labeling in panels b and c is very small. Suggest different arrangement of subplots to improve visibility and clarity.

- Table S1: What is the reasoning for the differences in analysis period for EN +SIOD and EN-only? Are the results robust to using the same time period?

REVIEWER COMMENTS:

Reviewer #1:

This manuscript is about the predictability of ENSO and the possible changes of the key-predictors of El Nino and La Nina events in the two last decades.

As noted already in the literature, they first illustrate that the traditional ENSO precursors, the Warm Water Volume (WWV) in the Pacific and the Indian Ocean (IO) tropical modes (e.g. the IOD and IOB modes) lose their significance since the 2000s as their relationships with the low-level zonal wind anomalies over the western Pacific have weakened significantly recently. The authors suggest that this evolution is related to the emergence of a new mode of variability in the south subtropical IO, the Southern Indian Ocean Dipole (hereafter SIOD), which is distinct from the tropical IO modes and the Indian Ocean Subtropical Dipole (IOSD hereafter) located more to the South (Behera and Yamagata 2001), which have already been largely studied in the literature and are known to be both partly forced by ENSO and possible ENSO precursors. Based on statistical analysis and different climate model experiments (with both coupled and atmospheric forced climate models), they suggest that the SIOD is now an active player in ENSO evolution and a key precursor of ENSO. However, the results and arguments presented in the manuscript fall short to be convincing to me for several reasons:

[Reply] We appreciate the reviewer for careful reading with priceless comments. We have revised the manuscript to fully resolve the reviewer's concerns by performing additional observational analysis and idealized model experiments. In addition, many parts of the earlier version of manuscript have been rewritten to satisfy the reviewer's request.

Major comments

1. First, the definition of a new climate (e.g., SST here) index requires the demonstration that this time series represents a coherent physical mode of variability or at least a mode, which represents a substantial part of the SST variability in the (Indian) domain. Obviously, the SST variability in the tropical IO is dominated by the IOD (and the IOB) and the IOSD in the subtropics and they are both intimately related, even without ENSO forcing, to sustain biennial variability in the IO (Cretat et al. 2018). The IOD and IOSD peak, respectively, in boreal fall (September and October) and boreal winter (e.g. from December to February), and the authors argue (without any justification) that their SIOD peaks between boreal fall and boreal winter, which corresponds to a season with low SST variability relative to boreal fall or winter in the IO. Nothing has been presented in the manuscript to justify that the SIOD is not an artificial mode and that its peak season is between the IOD and IOSD peak seasons. Furthermore, the existence of such mode can be hardly justified in the presence of remote ENSO forcing and regional IOD and IOSD forcings without dedicated analysis, and such analysis is clearly missing here.

My interpretation is that its definition has been mainly based to maximize its correlation with ENSO indices since the 2000s. This can be a start, but obviously this must be complemented by convincing arguments that the defined mode is a physical, not artificial, entity and the resulting relationship is not only the product of internal variability.

[Reply] Thank you for the reviewer's comment. To address the reviewer's comments, we firstly conducted an empirical orthogonal function (EOF) analysis of the SST anomalies during the OND season to show the SIOD mode is a dominant physical mode after 2000s (Figure 1-A). The 1st EOF is associated with the Indian Ocean basin-wide warming, which is not much changed between two decades (i.e., 1976-1997 and 1998-2019). On the other hand, the 2nd EOF exhibits dramatic differences in their spatial distribution. In particular, the 2nd EOF during 1998-2019 shows a clear dipole pattern over the southern Indian Ocean as shown in SIOD-related regression (Figure 1c of the main manuscript). The location of the negative, and positive SST anomalies in the 2nd EOF during 1998-2019 is mostly overlapped to [90°–120° E, 30°–5° S], and [65°–85° E, 25°–10° S], respectively, which are the regions to define the SIOD index. In addition, the seasonal standard deviation of the SIOD index during 1998-2019 shows a largest value during the OND season, which justifies our selection of the season to define the SIOD index (Figure 1-B).

The aforementioned findings are described in the revised manuscript as follows, and the Figs. 1-A and 1-B are added as Supp. Figs. 2 and 4, respectively.

Line 67-78: “The SIOD emerges from the southern IO during the October-November-December (OND) season during 1998-2019 as a 2nd empirical orthogonal function (EOF) of the SST anomalies over the Indian Ocean (Fig. S2d), differing from the Indian Ocean Subtropical Dipole (IOSD)(Behera and Yamagata, 2001; Morioka et al., 2013) which is appeared as a 2nd EOF during 1976-1997 (Fig. S2b). Based on the SIOD-related eigen-vector (i.e., 2nd EOF during 1998-2019), we define the SIOD index by taking SST anomaly differences between the southcentral IO [65°–85° E, 25°–10° S] and southeastern IO [90°–120° E, 30°–5° S] (shown in boxes of Fig. 1c). Note that the temporal lead-lag correlation between the SIOD and IOSD is weak at any lead/lag within 12 months, indicating that the SIOD is an independent mode to the IOSD (Fig. S3). Further, the SIOD index exhibits a peak intensity during the OND season (Fig. S4a), indicating that the peak amplitude is systematically delayed, and ahead in time than that of the IOD, and IOSD, respectively.”

Figure 1-A. (a) The first EOF of OND SST anomaly in the Indian Ocean during 1976-1997. (b) Same as (a) except for the second EOF. (c,d) Same as (a,b) except for during 1998-2019. The number in a parenthesis in a-d indicates the explained variance of each EOF mode. Units are non-dimensional.

Figure 1-B. The seasonal standard deviation of the SIOD index during 1998-2019.

2. Second, the results are largely overemphasized by the authors, as the SIOD seems only related to Niño4 SST and not to Niño3 or Niño3.4 SST, which are more traditional ENSO indices (see Fig. 3f). This also leads to the alternate hypothesis that SST variability in the central and west Pacific, represented by Niño4 SSTs, drives the SST gradient in the South IO related to the SIOD during recent decades, rather than the reverse (see Fig. 1c). This hypothesis is never explored or refuted in the manuscript. Such possibility can be checked by examining the correlations between the SIOD and SSTs at both positive and negative lags and by examining the persistence of Niño4 and Niño3 SSTs separately in Fig. 1f, which is fully justified as shown in Fig. 3f (rather than using Niño3.4 SSTs). In other words, the authors must demonstrate that the persistence of Niño4 SSTs by itself does not perform better than their SIOD for predicting Niño4 SSTs itself since the 2000s in observations.

[Reply] We highly appreciate for the reviewer's comment. We agree that the role of the SIOD is particularly robust for the Niño4 SST. To clarify this point, we have changed the title of this study to "Southern Indian Ocean Dipole as a trigger for **Central Pacific** El Niño since the 2000s". To demonstrate the role of the SIOD on the Niño4 SST, a lead-lag correlation between the SIOD during the OND season and the Niño4 SST is shown in Figure 1-C. Interestingly, the result shows two positive peaks at zero lag (i.e., the OND(0) season) and 13 months lag (i.e., the ND(+1)J(+2) season). The physical interpretation for the first peak has a causality issue as it occurs simultaneously during the OND season: it is hard to distinguish the role of the SIOD on the Niño4 SST from the vice versa only with the observational analysis as the reviewer pointed out.

On the other hand, as the second peak indicates that the SIOD leads the Niño4 SST for about a year (i.e., 13 months), it is highly probable that the SIOD is responsible for the Niño4 SST variability rather than vice versa. To support that the SIOD is responsible for the second peak, we additionally calculated the auto-correlation of the Niño4 SST (black line in Figure 1-C). It is clear that the second peak is not shown at all in the auto-correlation, which implies that the Niño4 SST is hardly sustained for two years. This clearly demonstrates that the positive SIOD during the OND season is responsible for the positive Niño4 SST anomaly with 13 months lag.

Aforementioned points are emphasized in the revised manuscript as follows:

Line 123-135: "The lead-lag correlation between the SIOD during the OND season and the Niño4 (SST averaged over 160° E – 150° W, 5° S – 5° N) shows two positive peaks at zero (i.e., OND(0)) and 13 months lag (i.e., ND(+1)J(+2)) (red line in Fig. 1f), while the correlation coefficients between the SIOD and Niño3 (SST averaged over 150° – 90° W, 5° S – 5° N) varies from about 0.13 to 0.48 which are not statistically significant (blue line in Fig. 1f). Here, a starting year is marked by (0) and a following year is marked by (+1). It should be noted that

the auto-correlation of the Niño3 and Niño4 gradually decay during the following year and eventually transit into a neutral state in the subsequent year's winter (gray and black lines in Fig. 1f), indicating that the Niño3 and Niño4 SST by itself is hard to persist for two consecutive years. This clearly demonstrates that the SIOD significantly contributes to the multi-year persistence of the SST anomalies over the equatorial central Pacific since 2000s. That is, the SIOD is a firstly explored precursor to prolong the ENSO signal, while the revealed precursors outside the tropical Pacific contribute to accelerate the ENSO phase transition (Chiang and Vimont, 2004; Izumo et al., 2010; Ham et al., 2013)"

Figure 1-C. Lead-lagged correlation coefficients between the SIOD index during the OND season and the Niño4 index (red), and auto-correlation coefficients of the Niño4 (black) from the JAS season to the subsequent year's NDJ season during 1998–2019. Open circles are statistically significant at the 95% confidence level.

3. Third, even if we admit that the SIOD is a coherent mode, the authors were not able to demonstrate clearly that something has changed around 2000 for explaining its emergence as an ENSO precursor. At the end of the manuscript (e.g., lines 193-221 and Figs. 4 and S8), the authors suggest that this is related to the westward shift of ENSO since the 2000s (see Fig. S7), which promotes colder SSTs in the eastern pole of their SIOD index. They justify this by showing differences between regressions with respect to Nino3 and Nino4 indices in observations (Fig. 4d) as well as differences between coupled experiments with Nino3 and Nino4 forcings, respectively (Fig. 4e), which both exhibit cold SST differences off Australia. However, my understanding is that Figs. 4d and e are simply the differences of the panels in each column of Fig. S8. Obviously, from Fig. S8, the differences between the Nino3 and Nino4 forcings over the IO are both very small and insignificant both for observations and model experiments. Moreover, the Nino4 forcing does not really produce cold SST anomalies off Australia (see Fig. S8d), so this argument does not hold. Furthermore, It is not obvious to me how these analyses can explain the decadal variability in the relationships between Nino4 SSTs and the SIOD at the interannual time scale, especially if the standard-deviation of the SIOD is the same before and after the 2000s as noted by the authors.

[Reply] We appreciate the reviewer for the valuable comment. Even though we tried to show the impact of the westward shifted ENSO on the SIOD variability after 2000s by comparing the regressed SST anomalies with respect to Niño3 and Niño4 indices in the previous Figure 4 and Supp. Figure 8, the differences were not clear as two Niño indices are strongly correlated. Therefore, in the revised manuscript, we have replaced the previous regression analysis with the composite analysis which are commonly used to separate the EP and CP ENSO (Figure 1-D). We used N_{CT} and N_{WP} index to define two types of ENSO events followed by Ren and Jin, 2011:

$$\begin{cases} N_{CT} = N_3 - \alpha N_4 \\ N_{WP} = N_4 - \alpha N_3 \end{cases} \quad \alpha = \begin{cases} 2/5, & N_3 N_4 > 0 \\ 0, & \text{otherwise.} \end{cases}$$

Here, N_3 and N_4 denote Niño3 and Niño4 indices, respectively. First, the years in which the N_{CT} and N_{WP} index during the boreal winter is above (less) 0.5°C (-0.5°C) are defined. Of those years, the EP El Niño (La Niña) year is defined when the N_{CT} index is greater (less) than the N_{WP} index. After that, we calculated the difference in the composite between the EP El Niño and La Niña cases and then the values were divided by 2 to obtain a final EP ENSO pattern. The CP ENSO pattern is also obtained using the same method for EP ENSO except for that CP El Niño (La Niña) year is defined when the N_{WP} index is greater (less) than the N_{CT} index.

The difference between the EP and CP ENSO composite is much clear with the modified figures (i.e., Fig. 4d or Supp. Figs. 12a and 12b in the revised manuscript), and it becomes consistent with previous studies to separate two types of ENSO events (Kug et al., 2009; Ashok et al., 2009). More importantly, SST response over the IO is also prominent during the CP ENSO composite; the cold SST anomalies with the southerly wind anomalies are clearly shown off the west coast of Australia only in the CP ENSO composite (Figure 1-D(b)). On the other hand, in the EP ENSO composite, the dipole SST response over the southern IO is not clear (Figure 1-D(a)). This supports our notion that the westward shift of the ENSO action center during the CP ENSO events can lead stronger dipole SST signal over the southern IO.

Observation (EP, CP ENSO composite)

Figure 1-D. Composite of SST ($^{\circ}\text{C}$) and 925 hPa wind (vector, $\text{m}\cdot\text{s}^{-1}$) anomalies during the DJF season for (a) EP ENSO and (b) CP ENSO events during 1976-2019.

In addition, we redo the partial nudging experiments by prescribing the EP and CP ENSO-related SST anomalies over the equatorial Pacific: The EP and CP ENSO-related SST anomalies obtained from the composite analysis were nudged in the tropical Pacific ($120^{\circ}\text{E} - 80^{\circ}\text{W}$, $20^{\circ}\text{S} - 20^{\circ}\text{N}$) with the seasonally varying climatological SST during 1998-2019, whereas the other oceans were run freely (Exp_C_EPEN and Exp_C_CPEN). A total of 20 ensemble members with difference initial conditions were conducted for both experiments. Each experiments was run for 5 months from October (0) to February (+1). The model results support observation results that the dipole mode over the southern IO is robust only during the CP ENSO forcing (Figure 1-E). This supports the observational finding that the westward-shifted ENSO center contributes to induce SIOD event (Figure 1-E(b)).

CESM1 (Exp_C_EPEN, CPEN – Exp_C_CTRL)

Figure 1-E. Changes of SST ($^{\circ}\text{C}$) and 925 hPa wind (vector, $\text{m}\cdot\text{s}^{-1}$) anomalies in the (a) Exp_C_EPEN and (b) Exp_C_CPEN compared to the control experiments during the DJF season in the CGCM. SST forcing is obtained from EP ENSO and CP ENSO composite.

Finally, to address the reviewer's question about the decadal modulation of the SIOD intensity, we examined the ratio of the seasonal standard deviation of the SIOD during 1998-2019 to the SIOD during 1976-1997. The standard deviation of the SIOD during the OND season is about the same before and after 2000s (i.e., ratio is close to 1) as we already pointed out, however, the standard deviations of the SIOD during the DJF and JFM seasons are systematically enhanced after 2000s. This is consistent with the longer persistency of the SIOD event after 2000s, which induces the persistent wind forcing over the equatorial Pacific.

Figure 1-F. The ratio of the seasonal standard deviation of the SIOD index during 1998-2019 to the SIOD index during 1976-1997.

4. Finally, there are a lot of errors, confusion and missing details in the legends of the Figures and in the methods Section, which make a hard job for the reader to understand what is exactly plotted in the figures. To name just a few, it is not clear: when and how, the time series are detrended before the various regression analyses or how these regressions are precisely performed; if the EXPSIOD coupled experiment integrates a double SIOD forcing (as written in Table S2), which can be hardly justified, or a simple SIOD forcing has written in lines 262-280. The description of the coupled model experiments is also lacking important details, are these nudging experiments? What is the length of these experiments?

[Reply] Thank you for the reviewer's comments. We performed two partial nudging experiments using CGCM to investigate the impact of the SIOD on the subsequent year's ENSO during the boreal winter; In the first experiment, the positive SIOD-related SST anomalies were nudged in the southern IO ($65^{\circ} - 120^{\circ} \text{ E}$, $30^{\circ} - 5^{\circ} \text{ S}$) with the seasonally varying climatological SST obtained from ERSST v5 during 1998-2019, whereas the other oceans were run freely (Exp_C_SIOD). The second experiment is a control experiment, which is identical with the former experiment except for the observed climatological SST anomalies were nudged in the southern IO (Exp_C_CTRL). The nudging time scale was prescribed as one day so that the simulated SST state in the southern IO is strongly nudged toward a given state of the observed SST. For obtaining the positive SIOD-related SST anomaly pattern, we first collected the years in which the OND SIOD index is larger (less) than one standard deviation for the composite analysis. Of those years, we calculated the difference in the composite between the positive SIOD and negative SIOD events and then the values were divided by 2 to obtain a final SIOD-related SST anomaly pattern. Finally, the composited SST anomalies were added to the climatological SST field. A total of 20 ensemble members with different initial conditions were conducted for both experiments. Each experiment was run for

17 months from October (0) to February (+2).

We have also corrected the titles in the Figure legends and added some details about detrend method as follows:

“Figure 1 | Southern Indian Ocean Dipole as the precursor of the ENSO after 2000s.”

“Figure 2 | Atmospheric variances associated with the SIOD.”

“Figure 3 | Idealized coupled model experiments and correlation skill of the SIOD index as the ENSO precursor.”

“Figure 4 | The impact of the westward shifted ENSO on the SIOD variability after 2000s.”

Line 83-86: “Note that all the indices are detrended before the analysis by removing the linear trend within the each analyzed period (i.e., 1976-2019, 1976-1997, and 1998-2019) and the regression coefficients denote the changes with respect to the one standard deviation change of the indices.”

The legend of Fig. 4 is also particularly misleading as for the experiments what is presented is just differences between experiments and not differences between regressions as stipulated in the legend. Statistical significance of anomalies or differences is missing in many plots, which makes difficult to judge their importance (especially because many of them have small amplitudes), etc.

[Reply] We have corrected the legend of Figure 4e that this is the difference between the CGCM experiments, rather than between the regressions.

5. In summary, in my opinion, the authors defined their dipole index a priori and solely based on a lag correlation analysis of ENSO indices with SSTs before the ENSO peak (not shown in the paper). Otherwise, there is no proper physical justification that this climate index represents a well-defined physical phenomenon. amplitudes), etc.

[Reply] As a response to comment #1, we justified the usage of the SIOD index by demonstrating that the dipole mode over the southern IO is captured as the 2nd EOF of OND SST anomaly during 1998-2019.

Furthermore, what they call the « peak » season of this index corresponds to a minimum of SST variance in the Indian domain between the IOD and IOSD peak seasons.

[Reply] Thank you for pointing this out. To validate our choice for the season to define the SIOD index, we calculated the seasonal standard deviation of the SIOD, IOD, and IOSD index

during 1998-2019, respectively. The SIOD reaches its peak during the OND season (Figure 1-B), although the IOD, and IOSD exhibits their peak during the SON, and JFM season during 1998-2019 (Figure 1-G).

Figure 1-G. (a) The seasonal standard deviation of the IOD index during 1998-2019. (b) Same as (a) except for the IOSD index.

As the IOD and IOSD are significant precursors of each other, I guess that the SIOD has a strong statistical relationship with both the IOD and SIOD modes (not described here), but at present « the physics » of this index is not defined, at least not in the manuscript in its present form. etc.

[Reply] We appreciate the reviewer for the valuable comment. To address the reviewer's question about the relationship between the SIOD and IOD, or IOSD, we examined the temporal lead-lag relationship in Figure 1-H. The temporal correlation between the SIOD index and IOSD index is quite weak at any lead/lag within 12 months, indicating that the SIOD is an independent mode to the IOSD. We added this information in line 73-75 of the revised

manuscript.

On the other hand, the SIOD and IOD indices exhibit a significant temporal correlation from -3 to +3 months lag, as the northern loading of the SIOD-related SST anomalies is overlapped to the region to define the IOD index (Figs. 1-I(a) and 1-I(d)). Nevertheless, the systematic differences in the spatial distribution between the SIOD and IOD is also obvious; the meridional action center of the IOD-related SST anomalies is located around 5° S, while those of the SIOD is shifted to the south about 20 degrees. More importantly, their physical mechanism is distinct; the IOD is known to be coupled to the equatorial zonal wind anomalies (Figure 1-I(b)) (Saji et al., 1999), and captured as the 1st EOF during the SON season. On the other hand, SIOD is likely to be coupled to the meridional winds over the southern IO (Figure 1-I(f)), and captured as the 2nd EOF during the OND season (Figure 1-A). This demonstrate that the SIOD is a different physical mode from the IOD.

Figure 1-H. Monthly lead-lagged correlation coefficients between the SIOD-IOD index (blue line), and SIOD-IOSD index (black line) during 1998-2019. Open circles indicate correlation coefficients that are statistically significant at the 95% confidence level.

Figure 1-I. Regressions of (a) SST ($^{\circ}\text{C}$) and (b) 1000hPa zonal wind ($\text{m}\cdot\text{s}^{-1}$), and (c) 1000hPa meridional wind ($\text{m}\cdot\text{s}^{-1}$) anomalies during the SON season with respect to the normalized SON IOD index for 1998–2019. (d-f) Same as (a-c) except for the normalized OND SIOD index during the OND season. Shading denotes the region where the statistical significance is above the 95% (90% for U1000 and V1000) confidence level.

In addition, the lead relationship between the SIOD and Nino4 SSTs shown in the manuscript is not sufficient to convince me that the SIOD is not only just the result of the interplay between the remote ENSO forcing and regional forcings (e.g., the IOD and IOSD modes) and that this

lead relationship is not just a byproduct of internal variability as the period considered is very short (e.g., only 20 years).

[Reply] We do not exclude the possibility that the SIOD event can be generated by the combined impact of the simultaneous ENSO forcing and regional wind forcings. Our main point in this study is that, once the SIOD event is induced, it contributes to lead the Niño4 SST variation after a year. The lag-correlation supports our notion in two aspects: (1) the variation of the SIOD leads that of the Niño4 SST with a clear time-lags (i.e., a year), and (2) this significant lag relationship is not shown at all in the auto-correlation of the Niño4 SST (Figure 1-C). This demonstrates that the time-lag relationship between the SIOD and Niño4 SST is not a result of byproduct of the internal variability of the ENSO.

We agree with the reviewer's concern that the analyzed period is not sufficiently long. However, it was inevitable to utilize the limited time period (i.e., 22 years), as the SIOD became a dominant mode in the southern IO after 1998. That is, the relatively short analyzed period is mainly due to the strong decadal variation of the SIOD variability. We noted the limitation of the observational analysis originated by a relatively short analyzed period. We also noted that this limitation is quite common in the studies to examine the decadal variation of the interannual variabilities. The whole period is often divided into two or more number of times across the climate regime shifts (i.e., 1976/77, 1998/99) to examine the decadal variation of the interannual variabilities (Kwon et al., 2007; Chung and Li, 2013; Guo et al, 2016; Neske and McGregor, 2018; Li et al., 2020). To overcome the limitation caused by short analyzed period, we performed the idealized model experiments to support the observed role of the SIOD on the subsequent year's ENSO.

Other comments

1. Line 47: remove "that" after indicating, otherwise the sentence is incorrect.

[Reply] Corrected.

2. Lines 62-66: It is hard for me to see the logic here and how the "prominent background SST warming » lead to the definition of the new dipole index proposed here as the SIOD.

[Reply] Sorry for the confusion. We tried to mention that the atmospheric response to the given SST forcing over the IO is expected to be enhanced due to the stronger background SST warming during recent decades. As a reviewer pointed out, the increase of background SST is not directly related to the emergence of a new dipole mode over the southern IO. To avoid the

confusion, we deleted the corresponding sentence in the revised manuscript.

3. Lines 67-98 and Fig. 1: What is the logic and physical arguments leading to the definition of the “SIOD” index? This is not clear to me at present. Are the SST time series properly detrended before the correlation analysis? Contrary to what is argued (lines 89-90), the amplitude of the SST anomalies in the Pacific is not stronger with the “SIOD” index, only the correlations seem more significant

[Reply] As we stated as the response to comment #1, we provided the physical background of the SIOD index. The linear trend during the analyzed periods is removed before analysis, which is noted in the revised manuscript.

Sorry for the confusion about the word “stronger”. We tried to denote that the SST anomalies led by the SIOD event has a robust zonal action center over the equatorial central Pacific. To avoid the confusion, we modified this sentence as follows:

Line 105-106: “In particular, the zonal action center of the positive SST anomalies is located over the equatorial central Pacific, ...”

About the amplitude of the SST anomalies regressed onto the SIOD, it should be firstly noted that the El Niño led by the SIOD during the subsequent year’s boreal winter (Figure 1d in the main manuscript) is similar in magnitude (i.e., 0.5 °C over the equatorial Pacific) with that led by the IOD during 1977-1997 (Figure 1-J). This indicates that the influence of the SIOD on the SST anomalies over the equatorial central Pacific during the subsequent year is quantitatively comparable to that led by IOD. And, the amplitude of the SST anomalies in Figure 1 is simply regarded as the ENSO amplitude after a year of the SIOD event whose amplitude is equal to its one standard deviation (std). Therefore, for the SIOD events whose amplitude is larger than 1 std, the El Niño amplitude can be stronger than that in Figure 1. For example, the cases-averaged equatorial Pacific SST anomalies during the subsequent year of 2003 and 2008, when the SIOD index was greater than 1 std (i.e., SIOD during the OND season is 1.28 std and 1.04 std, respectively), reaches up to 1 °C (Figure 1-K). This is clearly over the criteria to be defined as the El Niño event.

Figure 1-J. Lagged regression of subsequent year's SST ($^{\circ}\text{C}$) anomalies during the DJF season with respect to the normalized SON IOD index for 1976-1997.

Figure 1-K. Composited SST ($^{\circ}\text{C}$) anomalies during the DJF season for the 2004 and 2009 El Niño events.

4. Furthermore, I will be very curious to see, if the Nino4 SST index taken in OND(0) will not lead to a better SST prediction in the tropical Pacific at one year lead than the "SIOD" index. In this respect, it will be more interesting to plot the respective lead-lag correlations with both Nino3 and Nino4 SSTs in Fig.1e and f as it is obvious from Fig. 3e and f, that the "SIOD" index is not a "significant predictor" of Nino3 or Nino34 SSTs, only Nino4 SSTs. By itself, Figs.1c and d can also be mainly due to the strong persistence of the Nino4 SSTs and their long lasting effects on the IO, rather than an indication of a real precursory signal in the IO! Moreover, the strong persistence of Nino4 SSTs after the 2000s is well documented in the literature, I think.

Lines 99-106: I think that these statements are again misleading as already noted above. First, Fig. 1f and the hypothesis that "SIOD" is a unique precursor of ENSO do not match with the results in Fig.3e-f, which show that both WWV and "SIOD" together fail to produce a useful prediction of both Nino3 and Nino34 SSTs. Taking into account the complex evolution of ENSO, the fact that SST (or any other variable) in some areas has a significant lead statistical

relationship with ENSO is not a proof by itself of a physically based precursory relationship. This has been illustrated many times in the literature, for example for the relationship between the Indian Monsoon and ENSO.

[Reply] Thank you for the comment. This is one of key points of our study that the forecast skill of Niño indices (particularly Niño4 index) is significantly improved when SIOD is used as a precursor. Figure 1-L exhibits the hindcast correlation skills of Niño indices with the various precursors. In Figure 1-L(a), the single precursor of the SIOD index during the OND season is used, and compared to those with Niño indices during the DJF season (Figure 1-L(c)). All the hindcast experiment is performed during 1998-2019. After roughly six months lags, the hindcast correlation skills of Niño3, Niño3.4, and Niño4 index with the precursor of the SIOD index are systematically superior to those with the Niño indices. Particularly, hindcast correlation skill of Niño4 is statistically significant up to 14 months lead, while that of Niño3 is not significant with 95% confidence level, which implies that the SIOD is a key precursor of CP ENSO after the 2000s. We emphasized this point in line 193-206 of the revised manuscript.

Furthermore, we constructed a multiple regression model to demonstrate the significant value added to El Niño prediction by including OHC influence due to the single precursor of the SIOD and Niño indices were not considered WWV effect which is an efficient predictor of the ENSO (Izumo et al., 2010). In Figure 1-L(b), the SIOD index during the OND season and ocean heat content (OHC) index during the DJF season are used (referred as SIOD+OHC), and compared to those with Niño indices and OHC index during the DJF season (referred as NINO+OHC) (Figure 1-L(d)). The hindcast correlation skills with the SIOD+OHC are also systematically superior to those with the NINO+OHC after a year (i.e., 13 months). This result demonstrates that the correlation skill of the Niño4 approached up to 0.5 with the SIOD+OHC is not due to the strong persistence of the Niño4. This is similarly shown in Figure 1f (or Figure 1-K in this rebuttal letter) that the Niño4 SST persists for two consecutive years with respect to the SIOD in the lag correlation, while it decays within a year in auto-correlation of the Niño4 SST. However, even though the forecast skills of Niño indices using the SIOD precursor are systematically higher than others, the improvement of the forecast skill of Niño3 is not robust as that of Niño3.4 and Niño4. This implies that the SIOD can be a powerful precursor of the CP ENSO event, and we noted this in the revised manuscript.

Figure 1-L. (a) Hindcast correlation skills of the Niño4 SST (red), Niño3.4 SST (grey), and Niño3 SST (blue) index from the OND to the subsequent year's NDJ season using the OND SIOD index during 1998-2019, respectively. (b) Same as (a) except for using the combination of the OND SIOD and DJF OHC (SIOD+OHC) index. (c,d) Same as (a,b) except for using the DJF NINO index and the combination of the DJF NINO and DJF OHC (NINO+OHC) index during 1998-2019. Grey boxes indicate the developing seasons of the SIOD. Open circles indicate correlation coefficients that are statistically significant at the 95% confidence level. A leave-one-year-out cross-validation method is applied.

5. Lines 122-141 and Fig. 2: How the ENSO signal has been removed? Only on the “SIOD” index on OND(0)? With what ENSO index: Niño3, Niño3.4 or Niño4 SSTs? Based on simultaneous covariability or with some lags? These details are critical and are missing here! Furthermore, the tight relationship with Niño4 SSTs during all seasons is clearly seen here, but difficult to see what is forcing what if the plots at negative lags are not shown.

[Reply] To avoid the possible confusion by removing the ENSO signal, we re-calculated Figs. 2a and 2b by simply plotting the simultaneous regressions of precipitation, 925hPa wind, and 925hPa streamfunction anomalies with respect to the OND SIOD index. That is, the ENSO signal is not removed in the new Figs. 2a and 2b. The cold SST anomalies over the southeastern IO reduce the precipitation over the maritime continent (Figs. 1c and 2a of the main manuscript). This negative precipitation with descending motion generates low-level divergent flow over

the maritime continent, inducing the westerly anomalies over the equatorial western Pacific (WP) (Figure 1-M(b)). However, as a reviewer pointed out, it is debating how much degree of the westerly anomalies over the equatorial western Pacific is contributed by the SIOD as the positive Niño4 SST anomalies is also shown in the regression.

To separate the role of the SIOD on the equatorial WP westerly from that of the Niño4 SST warming, we calculated the partial regression of equatorial (5° S – 5° N-averaged) 925hPa zonal wind (U925) anomaly over the Pacific during the following March-April-May (MAM(+1)) season onto the OND(0) SIOD index after excluding the impact of the D(0)JF(+1) Niño4 SST (160° E – 150° W, 5° S – 5° N) (red line in Figure 1-M(c)). Here, a starting year is marked by (0) and a following year is marked by (+1). For the comparison, the regressed MAM(+1) U925 anomaly onto the OND(0) SIOD index is also shown (blue line in Figure 1-M(c)). The partial regression exhibited similar amplitudes to that in the conventional regression over the equatorial WP, implying that the positive SIOD-related SST leads to westerly anomalies over the equatorial WP during the following MAM season without the SST forcings over the equatorial Pacific.

The role of the SIOD on the WP westerly anomaly is further supported by the partial lag-regression (Figure 1-M(d)). The WP westerly wind anomalies regressed onto the OND(0) SIOD index after removing the impact of the D(0)JF(+1) Niño4 SST gradually intensifies in time (blue line in Figure 1-M(d)), however, the WP westerly anomalies regressed onto the D(0)JF(+1) Niño4 SST after removing the impact of the OND(0) SIOD index decays (red line in Figure 1-M(d)). This shows that the WP westerly during the positive SIOD events shown in Figure 2b (or Figure 1-M(b)) is mostly led by the SIOD SST, rather than the equatorial Pacific SST anomalies. We added this information in line 145-160 in the revised manuscript.

Figure 1-M. Atmospheric variances associated with the SIOD. a-b, Regression maps of (a) precipitation ($\text{mm}\cdot\text{day}^{-1}$) and (b) 925hPa wind (vector, $\text{m}\cdot\text{s}^{-1}$), and streamfunction (contours, at intervals of $0.3\cdot 10^5 \text{ m}^2\cdot\text{s}^{-1}$) anomalies during the OND season with respect to the normalized OND SIOD index from 1998-2019. Shading and vector in a-b denote the region where the statistical significance is above the 95% confidence level. (c) Regression of 925hPa zonal wind anomalies averaged over the $5^\circ \text{ S} - 5^\circ \text{ N}$ over the Pacific during the following MAM season with respect to the normalized OND SIOD index for 1998-2019 (blue line). The red line is the same but for the partial regression after excluding the impact of the DJF Niño4 index. (d) Partial lag regression of 925hPa zonal wind anomalies averaged over the $120^\circ - 150^\circ \text{ E}$, $5^\circ \text{ S} - 5^\circ \text{ N}$ over the equatorial western Pacific from the OND season to the subsequent year's NDJ season with respect to the normalized OND SIOD index after excluding the impact of DJF Niño4 index from 1998-2019 (blue line). The red line is the same but for the partial lag regression with respect to the normalized DJF Niño4 after excluding the impact of the OND SIOD index for 1998-2019. Open circles in c-d indicate correlation coefficients that are statistically significant at the 95% confidence level.

6. As seen in Fig. 1c, the “SIOD” occurs mainly during ENSO, so the sentence “This implies that the atmosphere-ocean coupling processes sustain the cold anomalies over the west coast of Australia until the February-March-April (FMA) season when most El Niño events are initiated (Fig. 2e). » is difficult to justify here!

[Reply] Thank you for pointing this out. We removed the corresponding sentence in the revised manuscript.

7. Lines 154-165, Supplementary Table S1: Details are missing here, what are the years entering in the composites presented in Table S1? Authors must also compute if the statistics for the different terms in the mixed layer budget are really different for the SIOD+El Niño and El Niño cases, to demonstrate that the processes are really different between the two cases.

[Reply] Sorry for missing the corresponding information. We have added the details in the revised manuscript as below:

Line 168-173: “The El Niño events led by the SIOD (2002, 2003, 2004, 2009, and 2018) exhibit a weak contribution of the thermocline feedback (i.e., vertical advection led by mean upwelling and the anomalous temperature gradient) and a significant stronger contribution of the zonal advective feedback (i.e., zonal advection of mean SST by anomalous zonal currents) over the equatorial western-central Pacific than that of the El Niño events without the SIOD forcing (2006, 2014, and 2015) (Table. S1), ...”

In addition, as you pointed out, we computed the statistics for the different terms in the mixed layer budget and denoted the lower and upper bounds at the 95% confidence level based on the bootstrap method in the Supp. Table 1 in the revised manuscript.

8. Lines 166-176, Table S2 and CGCM experiments (lines 262-280): First, the description of the experiments lacks of details and is confusing. First, are these nudging experiments or the authors have simply inserted the SST forcing in each region in the coupling interface before sending the SST to the atmosphere? What is the length of these experiments, one year with 12 members each? The definition of the EXPSIOD is not the same in lines 262-280 and in Table S2 (where it is suggested that the SIOD forcing has been double), which one is the right one?

[Reply] Sorry for the confusion of the experiments lacks of details. The nudging scheme is one of simplest data assimilation techniques, which can be described as follows:

$$\frac{dSST}{dt} = A + \frac{(SST_{OBS} - SST)}{\tau}$$

, where SST, and A is the sea surface temperature in the CGCM, and the conventional dynamical and physical terms to lead SST tendency, respectively. Second term on the right-hand-side denotes the nudging term. By adding the nudging term, the simulated SST is always toward the observed SST. The nudging time scale (τ) is set to one day so that the simulated SST state in the southern IO was kept, more or less, in a state of the nudged SST. As we stated as the response to comment #4, the positive SIOD-related SST anomalies were nudged in the southern IO ($65^{\circ} - 120^{\circ}$ E, $30^{\circ} - 5^{\circ}$ S) with the seasonally varying climatological SST obtained from ERSST v5 during 1998-2019, whereas the other oceans were run freely (Exp_C_SIOD). The second experiment is a control experiment, which is identical with the former experiment except for the observed climatological SST anomalies were nudged in the southern IO (Exp_C_CTRL). The nudging time scale was prescribed as one day so that the simulated SST state in the southern IO is strongly nudged toward a given state of the observed SST. For obtaining the positive SIOD-related SST anomaly pattern, we first collected the years in which the OND SIOD index is larger (less) than one standard deviation for the composite analysis. Of those years, we calculated the difference in the composite between the positive SIOD and negative SIOD events and then the values were divided by 2 to obtain a final SIOD-related SST anomaly pattern. Finally, the composited SST anomalies were added to the climatological SST. A total of 20 ensemble members with different initial conditions were conducted for both experiments. Each experiment was run for 17 months from October (0) to February (+2).

Finally, while I fully agree that these experiments suggest that the south Indian Ocean SSTs can potentially force ENSO, the main effect in the EXPSIOD experiment is to enhance classical El Niño events (as with IOSD forcing during boreal spring), not CP El Niño events (e.g., SIOD forcing), which is radically different from what is seen in the observations, e.g. compare Figs. 3d and f. How the authors interpret or explain these differences? In this respect, it will be illuminating to see the panels corresponding to seasons between OND(0) and JJA(+1), which are presently not shown in Fig. 3, in order to understand how the classical El Niño are triggered in the EXPSIOD experiment as the westerly wind anomalies over the western Pacific are not significant during OND(0) (Fig. 3a). If the forcing in the EXPSIOD experiment has been double, this must also be clearly stated in the text and not only in Table S2.

[Reply] This is an important issue which should be addressed in more detail. As a reviewer pointed out, while the observed El Niño led by the SIOD is CP-type, the idealized experiments using the CGCM simulate mixed-type El Niño during the following boreal summer by given SIOD SST forcing. It is related to the systematic error of the current generation of CGCMs in simulating two different types of the El Niño events (Ham and Kug, 2012; Cai et al., 2018; Ren et al., 2019). In the hindcast experiments using the current generation of the forecast systems,

two types of El Niño events can only be distinguished only at 1-month lead forecast, and tends to be merged into a single type after two months lead forecasts (Ren et al., 2019). The CGCM used in this study (i.e., CESM1) also has difficulty in simulating independency between two types of El Niño events as shown in the observation (Cai et al., 2018). This indicates that the zonal SST action center during the CP-type El Niño tends to be extended to the east in most of CGCMs, and similar to those during the EP-type El Niño events.

9. Lines 177-186 and Figs 3.e-f: What is the rationale of using WWV during fall here? Please explain. Also what are the respective role of WWV and the “SIOD” during the second period? this is critical to demonstrate the key role of “SIOD” after the 2000s. Surprisingly, WWV does not lead to any skill full prediction of Nino3 or Nino3.4 SSTs during the first period as well, probably because WWV has the least persistence during OND(0), I guess. Please try to better illustrate the intrinsic role of SIOD in the ENSO skills! auto-correlations?

[Reply] We used the WWV index during the OND season as it is the latest season we can use for the hindcast experiment. For example, the WWV index during the DJF season cannot be used as the predictor as it is a future season after the forecast starts. Due to the same reason, Izumo et al. (2010), which emphasized the role of the IOD on predicting the subsequent year’s ENSO, used the WWV during the SON season, which is the season that the forecast starts. In addition, the hindcast results are almost same once we used OHC during the DJF season as in the response to minor comment #4 and Figure 1-L. This indicates that the changes in season to obtain OHC index do not affect our results.

10. Lines 188-192: Again, I will be interested to see the same plot as Fig. 4a for the Nino4 SST. Are the time series detrended before computing the auto-correlations?

[Reply] The auto-correlation of the Niño4 SST is given in Figure 1-N. As we highlighted earlier, the auto-correlation of the Niño4 SST, clearly demonstrated that the equatorial western-central Pacific warming totally decays after a year, and its occurrence for two consecutive years cannot be explained by its persistency. All the indices are detrended before the analysis by removing the linear trend during the each analyzed period (i.e., 1976-2019, 1976-1997, and 1998-2019).

Figure 1-N. Auto-lagged correlations of the Niño4 SST index from the DJF to subsequent year’s DJF season for 1976–1997 (gray) and 1998–2019 (black), respectively.

11. Lines 282-299, Atmospheric Global Climate Model (AGCM): Line 287, it is written that four experiments have been performed, but only two are described, please correct. Later it is written that the length of the simulations is one year, but finally it is stated that each experiment was integrated for 40 years, difficult to follow! Please correct ...

[Reply] Sorry for the confusion. We corrected the following sentences in the revised manuscript:

Line 320: “Two AGCM experiments were conducted: ...”

Line 327-329: “A total of 35 ensemble members with different initial conditions were conducted for both experiments. Each experiment was run for one year from October (0) to September (+1).”

12. Figure 2: Correct the legend for the statistical significance indicated by diagonal lines as presently we cannot know what variables are concerned (e.g., SST or heat content for the left panels, and rainfall or 925 hPa stream function for the right panels).

[Reply] We removed the diagonal lines and replaced the shading (i.e., ocean heat content, precipitation) and vectors (i.e., 925hPa wind) to denote the statistical significance in Figure 2 in the revised manuscript.

3. Supplementary Fig. 1a: The text in the panel suggests that the moving correlations are between IOD, IOB and the FMA zonal wind over the western Pacific instead of with Nino34 SST during NDJ as written in the legend, please correct.

[Reply] Sorry for the confusion. We corrected the legend in Supp. Figure 1a as below:

Figure 1-O. The 15-yr moving correlations between the IOD index during the SON season, the IOBM index during the DJF season, and the subsequent year's Niño3.4 index during the DJF season (blue line: IOD vs Niño3.4; red line: IOBM vs Niño3.4) for 1976-2019. The x axis indicates the middle year in the 15-yr moving winter (e.g., 1999 indicates the correlation coefficient from 1992-2006). Dots denote correlation coefficients that are statistical significance at the 95% confidence level.

14. Supplementary Figs. 3 and 4: Explain clearly how these panels have been obtained! Are these composite or regression maps? Here the data are detrended, which was not done before, why? Finally, include statistical significance in panels (a) and (b) to help the reader to concentrate on the key-points in each panel.

[Reply] Sorry for missing the corresponding information. The Supp. Figs. 3 and 4 are the result of the composite analysis. We have added the details in the legends of the figures and included statistical significance in the panels to help the potential reader to concentrate on the key-points as you mentioned in the revised manuscript with the modified figures (i.e., Supp. Figs. 7 and 8). In addition, we clearly stated that the data are detrended before the analysis by removing the liner trend during the entire analyzed period (1976-2019) in the revised manuscript.

15. Supplementary Fig. 8: The color bar is missing.

[Reply] Thank you for pointing this out. The color bar is added.

References

- Ashok, K., and T. Yamagata, 2009: Climate change: The El Niño with a difference. *Nature*, 461, 481-484.
- Behera, S. K., and T. Yamagata, 2001: Subtropical SST dipole events in the southern Indian Ocean. *Geophys. Res. Lett.*, 28, 327-330.
- Cai, W., and Coauthors, 2018: Increased variability of eastern Pacific El Niño under greenhouse warming. *Nature*, 564, 201-206.
- Chiang, J. C., and D. J. Vimont, 2004: Analogous Pacific and Atlantic meridional modes of tropical atmosphere–ocean variability. *J. Clim.*, 17, 4143-4158.
- Chung, P.-H., and T. Li, 2013: Interdecadal Relationship between the Mean State and El Niño Types. *J. Clim.*, 26, 361-379.
- Guo, Y., Z. Wen, and R. Wu, 2016: Interdecadal Change in the Tropical Pacific Precipitation Anomaly Pattern around the Late 1990s during Boreal Spring. *J. Clim.*, 29, 5979-5997.
- Ham, Y.-G., and J.-S. Kug, 2012: How well do current climate models simulate two types of El Niño? *Clim. Dynam.*, 39, 383-398.
- Ham, Y.-G., J.-S. Kug, J.-Y. Park, and F.-F. Jin, 2013: Sea surface temperature in the north tropical Atlantic as a trigger for El Niño/Southern Oscillation events. *Nature Geosci.*, 6, 112-116.
- Izumo, T., and Coauthors, 2010: Influence of the state of the Indian Ocean Dipole on the following year's El Niño. *Nature Geosci.*, 3, 168-172.
- Kug, J.-S., F.-F. Jin, and S.-I. An, 2009: Two types of El Niño events: cold tongue El Niño and warm pool El Niño. *J. Clim.*, 22, 1499-1515.
- Kwon, M., J. G. Jhun, and K. J. Ha, 2007: Decadal change in east Asian summer monsoon circulation in the mid-1990s. *Geophys. Res. Lett.*, 34.
- Li, X., Z. Z. Hu, B. Huang, and F. F. Jin, 2020: On the interdecadal variation of the warm water volume in the tropical Pacific around 1999/2000. *J. Geophys. Res. Atmos.*, 125, e2020JD033306.
- Morioka, Y., T. Tozuka, and T. Yamagata, 2013: How is the Indian Ocean subtropical dipole excited? *Clim. Dyn.*, 41, 1955-1968.
- Neske, S., and S. McGregor, 2018: Understanding the warm water volume precursor of ENSO events and its interdecadal variation. *Geophys. Res. Lett.*, 45, 1577-1585.
- Ren, H.-L., J. Zuo, and Y. Deng, 2019: Statistical predictability of Niño indices for two types of ENSO. *Clim. Dyn.*, 52, 5361-5382.
- Ren, H. L., and F. F. Jin, 2011: Niño indices for two types of ENSO. *Geophys. Res. Lett.*, 38.
- Saji, N., B. N. Goswami, P. Vinayachandran, and T. Yamagata, 1999: A dipole mode in the tropical Indian Ocean. *Nature*, 401, 360-363.

REVIEWER COMMENTS:

Reviewer #2:

Using the observations and coupled model results, the authors have discussed the inter-basin interaction between the Indian Ocean and the Pacific with focusing on three climate phenomena, i.e., IOD, ENSO and Southern Indian Ocean Dipole (SIOD). Since the interaction among climate modes is becoming a hot topic in the climate research community, I have read through this manuscript with much interest. They claim that the SIOD is becoming a key precursor of El Niño since the 2000s at a 13-month lead. They also claim that not only has the leading role influencing ENSO changed from IOD to SIOD, but the role itself has changed; IOD accelerates the transition of ENSO, while the SIOD prolongs it for two consecutive years.

[Reply] Thank you for the careful reading with priceless comments. We have addressed all issues raised by the reviewer. The manuscript has been significantly revised according to the reviewer's comments.

Major comments

1. Although the conclusion was obtained by rather mediocre analyses, it seems reasonable. My main criticism is that the authors do not properly assess the closely related literature done so far in a somewhat different context. The phenomenon that the authors call SIOD has already been studied intensively as Ningaloo Niño, and the atmospheric (and even oceanic) inter-basin interaction has also been studied in detail. Also, there is much research on the ENSO diversity (Modoki or Dateline/CP El Niño, whatever authors may call) since Ashok et al. (2007)' detailed work. In this sense, the present work related to those aspects may be classified as a review work. A novel point may be the relationship between SIOD and CP El Niño in the Pacific since the 2000s. Therefore, the storyline needs to be re-written, together with giving credit to the precedent literature in a suitable way rather than following the process of the authors' appreciation of the problem. It will be necessary to comment on the relationship between the negative SIOD (Ningaloo Niña) and La Nina Modoki. Although it was in 1982/83 before the authors' regime shift, possible comments on the unusual case of coexistence of positive SIOD (Ningaloo Niño) and El Niño seems to inspire readers.

[Reply] We appreciate the reviewer's constructive comments. We totally agree with that the references mentioned by the reviewer are closely related to our study, therefore, it is worthwhile to be referred in the revised manuscript as follows:

Line 60-62: "On the other hand, the Ningaloo Niño in the southeastern Indian Ocean has been a marked increase in the occurrence, which can influence the tropical Pacific variability since the late 1990s(Kataoka et al., 2014; Feng et al., 2015; Zhang and Han, 2018)."

As there is a spatial similarity between the Ningaloo Niña and positive SIOD event, their different role on the subsequent year's ENSO should be addressed. Figure 2-A shows the lag

regression between the Ningaloo Niño index during the OND season and the SST anomalies during the next year's boreal winter for 1998-2019. While the SIOD induces the subsequent year's ENSO event with a significant amplitude, the equatorial Pacific SST anomalies led by the Ningaloo Niño is negligible at the 95% confidence level. In addition, the ENSO response is weaker in case of Ningaloo Niña even though its negative SST anomalies over the southeastern IO is almost twice to that during the SIOD event. This indicates that the dipole SST pattern associated with the SIOD, rather than the uni-polar SST associated with the Ningaloo Niña, plays a stronger role to lead the ENSO. Therefore, the SIOD is a much powerful index than the Ningaloo Niño index to be linked to the ENSO during a subsequent year. We have added those discussion in line 121-122 and the Supp. Figure 5 in the revised manuscript.

Furthermore, we have commented the unusual case as the reviewer suggested in line 99-101 of the revised manuscript as follows: "This simultaneous relationship is quite rigorous in individual cases even though few exceptional cases of coexistence of the positive SIOD and La Niña events (i.e., 2008, 2017) also exist."

Figure 2-A. Lag regressions of (a) SST (°C) anomalies during the OND season and (b) subsequent year's SST (°C) anomalies during the DJF season with respect to the normalized OND Ningaloo Niño index from 1998–2019. We multiplied -1 to the regressed field to match the sign of Ningaloo Niño to the eastern loading of the SIOD event. Shading denotes the region where the statistical significance is above the 95% confidence level.

Minor comments

1. L. 42-52: ENSO diversity should be touched on at an early stage somewhere here

[Reply] As we stated as the response to comment #1, we have added the sentences related to ENSO diversity in the revised manuscript as follows:

Line 51-53: “In addition, ENSO behaviors have become increasingly diverse and therefore difficult to predict ENSO events since the 2000s(Ashok et al., 2007; Kug et al., 2009; Barnston et al., 2012).”

2. L. 63-66, L. 107-112: Some readers might think SIOD stands for Subtropical Indian Ocean Dipole as introduced by Behera and Yamagata (2001).

See https://en.wikipedia.org/wiki/Subtropical_Indian_Ocean_Dipole

[Reply] We totally agree with your opinion. To show that the Southern IOD mode used in this study differs from the Subtropical IOD in Behera and Yamagata (2001), we conducted an empirical orthogonal function (EOF) analysis of the SST anomalies during the OND season (Figure 2-B). The 1st EOF is associated with the Indian Ocean basin-wide warming, which is not much changed between two decades (i.e., 1976-1997 and 1998-2019). On the other hand, the 2nd EOF exhibits dramatic difference in their spatial distribution. In particular, while the 2nd EOF during 1998-2019 shows an east-west oriented SST dipole pattern over the southern Indian Ocean (i.e., Southern IOD mode), a northeast-southwest SST dipole pattern (i.e., Subtropical IOD mode) is exhibited during 1976-1997. This implies that the Southern IOD mode suggested in this study differs from the Subtropical IOD mode as introduced by Behera and Yamagata (2001) in terms of the maximum SST location.

The aforementioned discussions are added in the revised manuscript as follows. In addition, the Figure 2-B is added as Supp. Fig. 2. Even though the SIOD might cause the confusion by referring both Southern IOD and Subtropical IOD, we keep our original notation (i.e., SIOD) as the subtropical IOD is often called SEIO SST (Dominiak and Terray, 2005), or IOSD (Morioka, et al., 2013).

Line 67-73: “The SIOD emerges from the southern IO during the October-November-December (OND) season during 1998-2019 as a 2nd empirical orthogonal function (EOF) of the SST anomalies over the Indian Ocean (Fig. S2d), differing from the Indian Ocean Subtropical Dipole (IOSD)(Behera and Yamagata, 2001; Morioka et al., 2013) which is appeared as a 2nd EOF during 1976-1997 (Fig. S2b). Based on the SIOD-related eigen-vector (i.e., 2nd EOF during 1998-2019), we define the SIOD index by taking SST anomaly differences between the southcentral IO [65°–85° E, 25°–10° S] and southeastern IO [90°–120° E, 30°–5° S] (shown in boxes of Fig. 1c).”

Figure 2-B. (a) The first EOF of OND SST anomaly in the Indian Ocean during 1976-1997. (b) Same as (a) except for the second EOF. (c,d) Same as (a,b) except for during 1998-2019. The number in a parenthesis in a-d indicates the explained variance of each EOF mode. Units are non-dimensional.

3. L. 76-78: Some positive (negative) IODs co-occur with La Niña (El Niño)!

[Reply] Thank you for pointing this out. We have corrected the sentence in the revised manuscript as follows:

Line 89-91: “Furthermore, positive SST anomalies are observed over the equatorial central-eastern Pacific, indicating that the positive IOD events tend to co-occur with the El Niño events(Webster et al., 1999).”

4. L. 82-91: The statistical correlation has nothing to do with the causality in general. We must be careful in wording.

[Reply] Thank you for the valuable comment. We acknowledge that the regression analysis cannot explain causality between variables. To address reviewer's comment, we have removed the associated verbs in line 85 and 90 in the manuscript and added new verbs to avoid the confusion in the revised manuscript as follows:

Line 96: "... the positive SIOD events tend to co-occur ..."

Line 104: "... the positive SIOD events have a chance to induce ..."

5. L. 101: red line \diamond blue line

[Reply] Thank you for pointing this out. We corrected it.

6. L. 103: The clear identification of the ENSO diversity was first done by Ashok et al. (2007). Credit should not go to follow-on works

[Reply] Thank you very much for proving related the reference. Ashok et al. (2007) is cited in the revised manuscript.

7. L. 107-112: If the authors claim a new SIOD climate mode as an east-west dipole, this simple description is not enough. As I mentioned already, the anomaly west of the Australia is already known as Ningaloo Niño (Niña) and catalogued as a new climate mode first by Kataoka et al. (2013)

Ref: On the Ningaloo Nino/Nina. 2013, Climate Dynamics. Doi:10.1007/s00382-013-1961-z.

[Reply] The spatial similarity and differences between the SIOD and the Ningaloo Niño (Niña) is fully recognized in line 114-122 of the revised manuscript, and Kataoka et al. (2014) is added as a reference. More importantly, as we stated as the response to comment #1, we provided the evidence of the difference role of the SIOD from that of the Ningaloo Niño index on the ENSO.

8. L. 99: Authors should recognize the case in which Ningaloo Niño occurred with El Niño in 1982/83.

[Reply] Thank you for pointing this out. We have added the reference of Kataoka et al. (2014) in the corresponding sentence.

9. L.99-141: Authors should know Tozuka et al. (Climate Dynamics, 2014: doi:10.1007/s00382-013-29044-x). Almost everything here was already discussed there.

[Reply] Thank you very much for proving the related reference. We acknowledge that atmospheric circulation anomalies of the SIOD-related SST forcing are similar to that of the

Ningaloo Niño/Niña-related SST forcing as shown in Tozuka et al. (2014). This information is added in line 192 of the revised manuscript.

10. L.156: Is Table S1 actually derived by the surface mixed-layer heat budget analysis? What happened to the ocean diabatic processes? Also, why the total percentage exceeds 100%?

[Reply] Sorry for the confusion about the percentages in Table S1. The percentages denote the ratio of the temperature tendency to each term (advection terms). We have added the details in the caption of Supp. Table 1.

11. L.173-174: This is exactly what Tozuka et al (2014) did.

[Reply] Thank you very much for proving the related reference. We acknowledge that the design of AGCM simulation in this study is very similar with that of the AGCM simulations as introduced Tozuka et al. (2014).

12. L.198-L.221: I could not follow the meaning of Nino4 index and Nino3 index in this paragraph. It appears those are different from the conventional usage.

[Reply] Sorry for the confusion. To avoid the confusion, we replaced the previous Supp. Figure 7 by following the definition of the EP and CP ENSO in previous study (Ren and Jin, 2011). We used N_{CT} and N_{WP} index to define two types of ENSO events as follows:

$$\begin{cases} N_{CT} = N_3 - \alpha N_4 \\ N_{WP} = N_4 - \alpha N_3 \end{cases} \quad \alpha = \begin{cases} 2/5, & N_3 N_4 > 0 \\ 0, & \text{otherwise.} \end{cases}$$

Here, N_3 and N_4 denote Niño3 and Niño4 indices, respectively. First, the years in which the N_{CT} and N_{WP} index during the boreal winter is above (less) 0.5°C (-0.5°C) are defined. Of those years, the EP El Niño (La Niña) year is defined when the N_{CT} index is greater (less) than the N_{WP} index. After that, we calculated the difference in the composite between the EP El Niño and La Niña cases and then the values were divided by 2 to obtain a final EP ENSO pattern. The CP ENSO pattern is also obtained using the same method for EP ENSO except for that CP El Niño (La Niña) year is defined when the N_{WP} index is greater (less) than the N_{CT} index. Information is added in line 305-313 of the revised manuscript.

13. L.203-L.212: I could not follow the description here. Figures look almost the same.

[Reply] As a response to minor comment #12, Even though we tried to show the impact of the westward shifted ENSO on the SIOD variability after 2000s by comparing the regressed SST

anomalies with respect to Niño3 and Niño4 indices in the previous Figure 4 and Supp. Figure 8, the differences were not clear as two Niño indices are strongly correlated. Therefore, in the revised manuscript, we have replaced the previous regression analysis with the composite analysis which are commonly used to separate the EP and CP ENSO (Ren and Jin, 2011).

The difference between the EP and CP ENSO composite is much clear with the modified figures (i.e., Fig. 4d or Supp. Figs. 12a and 12b in the revised manuscript), and it becomes consistent with previous studies to separate two types of ENSO events (Kug et al., 2009; Ashok et al., 2009).

14. Figure 1: The title is strange from a logical viewpoint. The word suggests the existence of causality. Figures here just denote correlation. (f), -> <red>.

[Reply] Thank you for pointing this out. We have corrected the title to “Southern Indian Ocean Dipole as the precursor of the ENSO after 2000s”.

15. Figure 4: <c, Same as a> -> <c, Same as b>. <e, Same as c> -> <c, Same as d>.

[Reply] Thank you for pointing this out. It is corrected as a reviewer suggested.

References

- Ashok, K., and T. Yamagata, 2009: Climate change: The El Niño with a difference. *Nature*, 461, 481-484.
- Ashok, K., S. K. Behera, S. A. Rao, H. Weng, and T. Yamagata, 2007: El Niño Modoki and its possible teleconnection. *J. Geophys. Res. Ocean.*, 112.
- Barnston, A. G., M. K. Tippett, M. L. L'Heureux, S. Li, and D. G. DeWitt, 2012: Skill of real-time seasonal ENSO model predictions during 2002–11: Is our capability increasing? *Bull. Am. Meteorol. Soc.*, 93, 631-651.
- Behera, S. K., and T. Yamagata, 2001: Subtropical SST dipole events in the southern Indian Ocean. *Geophys. Res. Lett.*, 28, 327-330.
- Dominiak, S., and P. Terray, 2005: Improvement of ENSO prediction using a linear regression model with a southern Indian Ocean sea surface temperature predictor. *Geophys. Res. Lett.*, 32.
- Feng, M., and Coauthors, 2015: Decadal increase in Ningaloo Niño since the late 1990s. *Geophys. Res. Lett.*, 42, 104-112.
- Kataoka, T., T. Tozuka, S. Behera, and T. Yamagata, 2014: On the Ningaloo Niño/Niña. *Clim. Dynam.*, 43, 1463-1482.
- Kug, J.-S., F.-F. Jin, and S.-I. An, 2009: Two types of El Niño events: cold tongue El Niño and warm pool El Niño. *J. Clim.*, 22, 1499-1515.
- Morioka, Y., T. Tozuka, and T. Yamagata, 2013: How is the Indian Ocean subtropical dipole excited? *Clim. Dyn.*, 41, 1955-1968.
- Ren, H. L., and F. F. Jin, 2011: Niño indices for two types of ENSO. *Geophys. Res. Lett.*, 38.
- Tozuka, T., T. Kataoka, and T. Yamagata, 2014: Locally and remotely forced atmospheric circulation anomalies of Ningaloo Niño/Niña. *Clim. Dynam.*, 43, 2197-2205.

Webster, P. J., A. M. Moore, J. P. Loschnigg, and R. R. Leben, 1999: Coupled ocean-atmosphere dynamics in the Indian Ocean during 1997-98. *Nature*, 401, 356.

Zhang, L., and W. Han, 2018: Impact of Ningaloo Niño on tropical Pacific and an interbasin coupling mechanism. *Geophys. Res. Lett.*, 45, 11,300-311,309.

REVIEWER COMMENTS:

Reviewer #3:

The study identifies subtropical Indian Ocean conditions as a skillful predictor for ENSO since the 2000s. Using observations, CGCM and AGCM experiments, the utility of the SIOD precursor is evaluated and potential mechanisms for the skill are detailed. Given ongoing challenges with skillful ENSO predictions in recent years, the study tackles a timely and relevant problem in ENSO forecasting. The paper is well-structured and clearly written overall. However, there are inconsistencies in the analyses in different parts of the study that need to be resolved and further clarification required on several aspects (see below comments).

[Reply] We appreciate the reviewer for careful reading with priceless comments. To satisfy the reviewer's comments, we have performed additional analysis and many parts of the earlier version of manuscript have rewritten. The revised manuscript also includes new sentences and figures to address reviewer's requests.

Main comments

1. Inconsistencies exist with regard to seasons, analysis periods etc. in different analyses in the main part of the manuscript and supplement. Or at least, deliberate choices for different seasons (e.g., SON in Fig 1a vs OND in Fig 1c, but then NDJ for both b and d; or different analysis periods for EN and EN+SIOD years in Table S1) must be better motivated and justified in the text.

[Reply] Thank you for the reviewer's comment. The different selected seasons between the IOD and the SIOD are based on their peak seasons; the IOD exhibits its peak during the SON season (Saji et al., 1999), while the SIOD reaches its peak during the OND season (Figure 3-A). We briefly mentioned this point in line 76-79 of the revised manuscript. And, to avoid the possible confusion, we used the same analyzed periods (i.e., 1998-2019) for the analysis in Figure 1a-d and Supp. Table 1 of the revised manuscript. The main conclusion for Table S1 is still rigorous with the analysis using 1998-2019.

Figure 3-A. (a) The seasonal standard deviation of the SIOD index, and (b) IOD index during 1998-2019.

2. It is unclear why the analysis period 1976-2019 was chosen; it does not coincide with limited data availability of the used datasets, as most go back further (though, the GODAE-based WWV is only available post-1980). Suggest using a longer analysis period or provide further justification for this start period. In fact, it appears that the analyses periods coincide with the mid-1970s climate shift in the Indo-Pacific and/or IPO phasing. Yet, no mention is made anywhere as to the implications these factors might have for the results obtained here, or for either ENSO predictability, ENSO diversity, interbasin connectivity etc. This is a missed opportunity to embed the results obtained in this study into a broader discussion of multi-decadal variability in the Indo-Pacific and implications for ENSO characteristics and predictive skill.

[Reply] We appreciate the reviewer for giving us valuable comment. As a reviewer suspected, the analyzed time period is selected after the climate regime shift at 1976-77 (An and Wang, 2000; Terray and Dominiak, 2005). Terray and Dominiak (2005) found that the lag-relationship between Indian Ocean (IO) SSTs and ENSO had experienced a remarkable change after 1976-77 regime shift. They argued that the climate variability over the IO is induced by the atmosphere-ocean coupled process within the IO after the 1976-77 regime shift, and it plays an active role in the transition phases of ENSO. In addition, An and Wang (2000) found that, due to the North Pacific climate shift in the mid-1970s, the period, amplitude, spatial structure, and temporal evolution of the El Niño notably changed. The aforementioned discussions are added

in the revised manuscript as follows:

Line 262-264: “The analysis period spans from 1976 to 2019 as the period, amplitude, spatial structure, and temporal evolution of the El Niño, and its relationship with the Indian Ocean SST notably changed after mid-1970s (An and Wang, 2000; Terray and Dominiak, 2005)”

Specific comments

1. The writing throughout the manuscript would benefit from editing by a native English speaker. Some specific corrections are indicated below, but that is not an exhaustive list.

[Reply] Responding to the reviewer’s comment. Substantial copy-editing was conducted by a native English speaker throughout the manuscript.

2. L33-34: For those unfamiliar with the IOD, the description of SIOD in the abstract is unhelpful. Suggest describing the SIOD pattern/anomalies in its own right without invoking another climate mode whose pattern then needs to be shifted latitudinally by a certain amount to achieve the overall pattern. This is even more apparent given that the next sentence refers to other factors (sign, timing etc) that differ between IOD and SIOD as precursors for ENSO.

[Reply] Thank you for pointing this out. We have removed the description of the SIOD by using the IOD and added the following sentence in the abstract.

Line 32-35: “Here we show that the Southern Indian Ocean Dipole (SIOD), which is characterized by east-west oriented sea surface temperature (SST) dipole pattern over the southern Indian Ocean, becomes a key precursor of the Central Pacific El Niño events since the 2000s at a 14-month lead.”

3. L36: It is unclear whether the “it” at the end of the sentence refers to ENSO or its transition. Please reword.

[Reply] Sorry for the confusion. The word “it” refers the period of the ENSO. We have corrected the corresponding sentence as below:

Line 35-37: “The role of the SIOD on the subsequent year’s ENSO is distinctive from the equatorial Indian Ocean Dipole mode(Saji et al., 1999) by prolonging the ENSO period.”

4. L47: A verb is missing in this sentence; “... lead-lag relationship exists between...”?

[Reply] It is corrected as a reviewer suggested.

5. L50: “less periodical” – this is vague and unclear. Does this refer to El Niño-La Niña transitions or periodicity between subsequent El Niño events or persistence of events? Please clarify.

[Reply] Sorry for the confusion. We tried to express that the transition of ENSO phase from one to the other (i.e., phase transition from El Niño to La Niña, or from La Niña to El Niño) has weakened since the 2000s. We have corrected the sentence as below:

Line 50-51: “..., resulting in the transition of ENSO phase from one to the other becomes less periodical compared to the period from 1979-1999.”

6. L67-69: Sentence is grammatically incorrect.

[Reply] Sorry for the confusion. We have corrected the sentence as below:

Line 79-80: “To demonstrate the newly emerged role of the SIOD on the subsequent year’s ENSO after the 2000s, ...”

7. L71: The boxes identified for the SIOD here differ significantly from earlier SIOD definitions. While this is acknowledged later in the manuscript, it would be helpful to provide some justification for the chosen regions here.

[Reply] Thank you for the reviewer’s comment. I guess the reviewer mentioned Subtropical IOD (https://en.wikipedia.org/wiki/Subtropical_Indian_Ocean_Dipole). By conducting an empirical orthogonal function (EOF) analysis of the SST anomalies during the OND season (Figure 3-B). in two period (i.e., 1976-1997 and 1998-2019), we found that the Subtropical IOD (i.e., earlier SIOD), and Southern IOD (i.e., SIOD in this study) is a second dominant mode during 1976-1997, and 1998-2019, respectively. The 2nd EOF during 1976-1997 shows a spatial distribution which is almost same with that in Subtropical IOD (Figure 3-C). On the other hand, the 2nd EOF during 1998-2019 shows a clear dipole pattern over the southern Indian Ocean as similarly shown in SIOD-related regression. The location of the negative, and positive SST anomalies in the 2nd EOF during 1998-2019 is mostly overlapped to [90°–120° E, 30°–5° S] and [65°–85° E, 25°–10° S], which are the regions to define the SIOD index, respectively. The aforementioned discussions are added in the revised manuscript as follows, and the Figure 3-B is added as Supp. Fig. 2.

Line 67-73: “The SIOD emerges from the southern IO during the October-November-December (OND) season during 1998-2019 as a 2nd empirical orthogonal function (EOF) of the SST anomalies over the Indian Ocean (Fig. S2d), differing from the Indian Ocean Subtropical Dipole (IOSD)(Behera and Yamagata, 2001; Morioka et al., 2013) which is appeared as a 2nd EOF during 1976-1997 (Fig. S2b). Based on the SIOD-related eigen-vector (i.e., 2nd EOF during 1998-2019), we define the SIOD index by taking SST anomaly differences between the southcentral IO [65°–85° E, 25°–10° S] and southeastern IO [90°–120° E, 30°–5° S] (shown in boxes of Fig. 1c).”

Figure 3-B. (a) The second EOF of OND SST anomaly in the Indian Ocean during 1976-1997. (b) Same as (a) except for during 1998-2019. The number in a parenthesis in a-b indicates the explained variance of each EOF mode.

Figure 3-C. Regressed Subtropical IOD index on detrended SST anomalies during 1958-2007. Adopted from https://en.wikipedia.org/wiki/Subtropical_Indian_Ocean_Dipole.

8. L74: Unclear why this analysis period 1976-2019 was chosen; see main comment above.

[Reply] As we stated as the response to comment #1, we have added the following sentence why we chose the analysis period of 1976-2019 in the revised manuscript.

Line 262-264: “The analysis period spans from 1976 to 2019 as the period, amplitude, spatial structure, and temporal evolution of the El Niño, and its relationship with the Indian Ocean SST notably changed after mid-1970s (An and Wang, 2000; Terray and Dominiak, 2005)”

9. L85: Reword to “... SIOD co-occurred...”

[Reply] Thank you for pointing this out. It is corrected as the reviewer suggested.

10. L89-91: Even more strikingly, the significant SST anomalies are not just located in the CP region (rather than EP), but at the poleward edges rather than along the equator. Further discussion of these poleward displaced features and their implications is required. What does the zonally broader SST anomalies imply for mechanisms/dynamics/feedbacks?

[Reply] Thank you for the valuable comment. The poleward extension of the positive SST anomalies might be related to the simultaneous relationship between the CP El Niño and the Pacific Meridional Mode (Stuecker, 2018). This possible mechanism is added in line 98-99 in the revised manuscript as follows: “The SST anomalies over the equatorial Pacific is extended poleward possibly due to fast positive feedback between the CP El Niño and the Pacific Meridional Mode(Stuecker, 2018).”

Through the oceanic temperature budget analysis, we found that the zonally broader SST warming, whose edge is extended up to the western Pacific, is caused by the zonal advective feedback. That is, as shown in Supplementary Table 1, the SST warming over the equatorial central Pacific is mainly led by the term $-u' \frac{\partial \bar{T}}{\partial x}$, which refers the eastward zonal current led by the westerly wind advects climatological warm SST over the western Pacific to the central Pacific (Kug et al., 2009).

11. L113: This is the first mention of the SEIO mode; please provide further explanation as to its pattern, mechanism and importance in the context here.

[Reply] Sorry for the possible confusion. The southeast Indian Ocean (SEIO) mode is characterized by cold SST anomalies during the February-March season in the southeast Indian Ocean, off the Australia coast. The core of this cold anomaly decreases slightly in intensity and area, moving northward from off the Australian coast to off the Java coast during spring and

summer, then off the Sumatra coast by October-November (Terray and Dominiak, 2005). Dominiak and Terray, (2005) argued that the SEIO SST anomalies during the late boreal winter can be a prominent precursor of ENSO. The description of the SEIO is briefly mentioned in the revised manuscript as follows:

Line 114-117: “The role of the SIOD on the development of the subsequent year’s ENSO is more pronounced than that of the SEIO (southeastern IO) mode(Terray and Dominiak, 2005; Dominiak and Terray, 2005), or Ningaloo Niño/Niña(Kataoka et al., 2014), which are characterized by cold SST anomalies during the February-March season in the southeast Indian Ocean, off the Australia.”

12. L118: Please cite p-values for the correlations

[Reply] Responding to the reviewer’s comment. We have cited p-values for the correlations in the revised manuscript.

13. L250: WWV data is not available for this time period

[Reply] Sorry for the unclear explanation. Originally, the WWV using the GODAS can only be calculated from 1980 onwards as the reviewer pointed out. Therefore, all the previous analysis (e.g., Figs. 2 and 3) using the WWV anomaly is the results by using the period after 1980. As other variables are used from 1976, this causes a serious misleading.

To match the period for the WWV to that of other variables, we replaced the WWV data from the GODAS to those from the Ocean Reanalysis System 5 (ORAS5) in the revised manuscript. Therefore, the results using WWV are updated from 1980 onwards in the original version to from 1976 onwards in the revised version. The results are not changed due to the modification of the WWV data product and period.

Figures & Supplementary material

14. Fig. 2: Areas indicating significant OHC are pretty small and disparate. Would they pass a field significance test? Same applies to precipitation.

[Reply] Thank you for the reviewer’s comment. The OHC anomaly always lags for at least several months behind the wind forcing due to its immense thermal inertial. So, the areas of significant OHC anomaly shown in Figure 2 are quite small and disparate. But, the significant

OHC areas gradually exhibits over the equatorial central Pacific during the following summer (Figure 3-D). As the OND(0) is the season when the SIOD-related OHC signal over the equatorial Pacific is bound to be small, we removed the OHC variable and focus on the atmospheric variables such as precipitation, 925hPa wind, and 925hPa streamfunction anomalies, which respond to the SST forcing instantaneously. In addition, in new Figure 2, the ENSO signal is not removed by using partial regression by avoiding excessively suppression of the climate variability over the equatorial Pacific.

Figure 3-D. Lag regression of ocean heat content (OHC) (contours, $10^{11} \text{ J}\cdot\text{m}^{-2}$) anomalies during the following year's boreal summer with respect to the normalized OND SIOD index from 1998-2019. Shading denotes the region where the statistical significance is above the 95% confidence level.

15. Fig. 3: What is the focus on JJA for the CGCM results? In the observations, NDJ (Fig. 1) or DJF/FMA (Fig. 2) was used.

[Reply] Thank you for the reviewer's comment. We tried to show the developing process of El Niño event during the following summer. As the following reviewer's suggestion, we have replaced the JJA season with the DJF season in Fig. 3c of the revised manuscript.

16. Fig. 4a-c: The legend text in grey is nearly invisible in a printed version; labeling in panels b and c is very small. Suggest different arrangement of subplots to improve visibility and clarity.

[Reply] Thank you for pointing this out. We enlarge the labels, and changed it color.

17. Table S1: What is the reasoning for the differences in analysis period for EN +SIOD and EN-only? Are the results robust to using the same time period?

[Reply] Responding to the reviewer's comment. We used previous decades to define EN-only case because a number of cases for EN-only during 1998-2019 is small as most of El Niño events are preceded by the SIOD after 1998. Note that this supports our notion that most of El Niño events after 1998 are led by the SIOD. To avoid the confusion, we have corrected it to use the same time period during 1998-2019 for both EN+SIOD and El-only. The main conclusion that the zonal advective feedback in EN+SIOD is systematically stronger than that in El-only is still rigorous.

References

- An, S.-I., and B. Wang, 2000: Interdecadal change of the structure of the ENSO mode and its impact on the ENSO frequency. *J. Clim.*, 13, 2044-2055.
- Behera, S. K., and T. Yamagata, 2001: Subtropical SST dipole events in the southern Indian Ocean. *Geophys. Res. Lett.*, 28, 327-330.
- Dominiak, S., and P. Terray, 2005: Improvement of ENSO prediction using a linear regression model with a southern Indian Ocean sea surface temperature predictor. *Geophys. Res. Lett.*, 32.
- Kataoka, T., T. Tozuka, S. Behera, and T. Yamagata, 2014: On the Ningaloo Niño/Niña. *Clim. Dynam.*, 43, 1463-1482.
- Kug, J.-S., F.-F. Jin, and S.-I. An, 2009: Two types of El Niño events: cold tongue El Niño and warm pool El Niño. *J. Clim.*, 22, 1499-1515.
- Morioka, Y., T. Tozuka, and T. Yamagata, 2013: How is the Indian Ocean subtropical dipole excited? *Clim. Dyn.*, 41, 1955-1968.
- Saji, N., B. N. Goswami, P. Vinayachandran, and T. Yamagata, 1999: A dipole mode in the tropical Indian Ocean. *Nature*, 401, 360-363.
- Stuecker, M. F., 2018: Revisiting the Pacific Meridional Mode. *Scientific reports*, 8, 3216.
- Terray, P., and S. Dominiak, 2005: Indian Ocean sea surface temperature and El Niño–Southern Oscillation: A new perspective. *J. Clim.*, 18, 1351-1368.

REVIEWER COMMENTS

Reviewer #2 (Remarks to the Author):

Re-review of "Southern Indian Ocean Dipole as a trigger for Central Pacific El Niño since the 2000s" entitled by Hyun-Su Jo et al.

The authors have responded to my first-round comments sincerely, particularly by crediting to antecedent important works. The main conclusion that the SST anomalies of the southeastern Indian Ocean off the west coast of Australia is becoming a key precursor of the recent westward-shifted El Niño (Dateline El Niño, Modoki El Niño, Central Pacific El Niño, whatever the authors may call) certainly deserves publication.

However, I still have serious doubts about the authors' introduction of the so-called SIOD. A solid physical analysis is required to claim a new climate mode; the linear mode expansion is inadequate. Well-accepted climate modes have been established after severe examination (e.g., <http://link.springer.com/article/10.1007/s00382-007-0356-4>). The so-called SIOD in this study is introduced only by use of a simple EOF analysis and has no solid physical background. It is necessary to demonstrate how it is formed both dynamically and thermodynamically and why it has a dipole structure. The authors' claim seems to be too easy.

Following the first-round criticism, the authors have included discussions about the Ningaloo Niña/Niño from l. 60 to l.67 as a kind of motivation of the present work. However, the statement from l. 124 to l. 126 is strange. Although Figure S5 looks almost the same with Fig. 1 c, d, the authors conclude that the Ningaloo Niña during the OND season on the subsequent year's ENSO is negligible! Readers might suspect that authors deliberately undermine the importance of SST anomalies related to the Ningaloo Niña in order to assert the importance of the so-called SIOD.

To confirm the influence of the SST anomaly of the southeastern Indian Ocean off the west coast of Australia, the authors have done the AGCM experiment which is almost the same with the Ref. 33 (Tozuka et al. 2014) and have reached the same conclusion (l. 199-l. 200). This again confirms that the SST anomalies in the Ningaloo Niño/Niña region plays a key role rather than does the dubious dipole structure of the so-called SIOD. This suggest that there is no need to introduce the SIOD as a new climate mode.

Based on the above, the authors should retreat from claiming the SIOD. Rather, they should focus on the importance of the SST anomalies in the Ningaloo Niño/Niña region. Even so, the punch line of the present work will be still alive as it is. Then, the title may be something like "Southeastern Indian Ocean SST anomalies as a trigger for Central Pacific El Niño since the 2000s" or simply "Ningaloo Niña as a trigger for Central Pacific El Niño since the 2000s".

Finally, I note that Ningaloo Nina events are more frequent in recent decades as partly seen in Table 1 of Tozuka et al. (2014). Also, Feng et al. (2021, J. Marine Systems, <https://doi.org/10.1016/j.jmarsys.2020.103473>) have recently discussed multi-year marine cold-spells off the west coast of Australia. Those seem to support the punchline of the authors.

Minor comment:

l.119: cold SST anomalies -> cold/warm SST anomalies

Reviewer #3 (Remarks to the Author):

The authors have mostly addressed previous concerns. While the response letter provides reasoning for the choices made regarding the analysis period (i.e., after the mid-1970s climate shift in the Indo-

Pacific) and a sentence to that effect has been added in L262-264, the broader issue has not been addressed: ie. the authors are encouraged to embed their results into a broader discussion of multi-decadal variability in the Indo-Pacific and implications for ENSO characteristics and predictive skill. This would greatly strengthen the study and raise its appeal to a broader readership. As it stands, the value is more narrowly focused on a readership with more discipline-specific interest in seasonal forecasting, as well as ENSO predictions and evolution.

The supplementary material provides an extensive set of additional information for interested readers. Given that Nature Communications allows for a larger number of display items as part of the main text, one consideration for the authors would be to add some of that material to the main part of the manuscript. This would improve the readability of the manuscript considerably, rather than referring extensively to the supplement.

The writing has improved to the previous version; however, the manuscript throughout would still benefit from further input from a native English speaker. Some specific examples for rewording are given below; however, this is not an exhaustive list.

Specific comments

- L50: Reword to "... and hence ENSO phase transitions becoming less regular..."
- L52: Reword to "... and it is therefore..."
- L60: Reword to "... has markedly increased in frequency, which..."
- L70: Reword to "... which appears as..."
- L78: That seems contradictory that SIOD peak amplitude is delayed and ahead in time? Please clarify.
- L117: Reword to "... off Australia..."
- L247: Should be "avenue" rather than "revenue"?
- Fig 1: Black boxes in a-d are hard to see/distinguish from other map features. Consider using a distinct color to better highlight the boxes.

REVIEWER COMMENTS:

Reviewer #2:

The authors have responded to my first-round comments sincerely, particularly by crediting to antecedent important works. The main conclusion that the SST anomalies of the southeastern Indian Ocean off the west coast of Australia is becoming a key precursor of the recent westward-shifted El Niño (Dateline El Niño, Modoki El Niño, Central Pacific El Niño, whatever the authors may call) certainly deserves publication.

However, I still have serious doubts about the authors' introduction of the so-called SIOD. A solid physical analysis is required to claim a new climate mode; the linear mode expansion is inadequate. Well-accepted climate modes have been established after severe examination (e.g., <http://link.springer.com/article/10.1007/s00382-007-0356-4>). The so-called SIOD in this study is introduced only by use of a simple EOF analysis and has no solid physical background. It is necessary to demonstrate how it is formed both dynamically and thermodynamically and why it has a dipole structure. The authors' claim seems to be too easy.

[Reply] Thank you for the careful reading with priceless comments. By performing the additional validation with a nonlinear statistical method (i.e., self-organizing maps (SOM)), we have confirmed that the SIOD is a physical mode. In addition, the physical mechanisms of the dipole structure in SIOD and its systematic difference from the Ningaloo Niña is provided in the revised manuscript by performing the oceanic mixed-layer heat budget analysis.

Major comments

1. Following the first-round criticism, the authors have included discussions about the Ningaloo Niña/Niño from l. 60 to l.67 as a kind of motivation of the present work. However, the statement from l. 124 to l. 126 is strange. Although Figure S5 looks almost the same with Fig. 1 c, d, the authors conclude that the Ningaloo Niña during the OND season on the subsequent year's ENSO is negligible! Readers might suspect that authors deliberately undermine the importance of SST anomalies related to the Ningaloo Niña in order to assert the importance of the so-called SIOD.

To confirm the influence of the SST anomaly of the southeastern Indian Ocean off the west coast of Australia, the authors have done the AGCM experiment which is almost the same with the Ref. 33 (Tozuka et al. 2014) and have reached the same conclusion (l. 199-l. 200). This again confirms that the SST anomalies in the Ningaloo Niño/Niña region plays a key role rather than does the dubious dipole structure of the so-called SIOD. This suggest that there is no need to introduce the SIOD as a new climate mode.

[Reply] We understand the reviewer's concern about the similarity between the Ningaloo Niña and the SIOD. The stronger impact of the SIOD on the subsequent year's ENSO than that of the Ningaloo Niña is due to the role of the western loading of the SIOD. To demonstrate this point, we calculated the partial regressions of SST anomalies during the OND(0) and the

following D(+1)JF(+2) season with respect to either the western, or eastern loading of OND(0) SIOD index after excluding the impact of the other (Figure 1-A).

The SST anomalies regressed on the western loading of the SIOD exhibit positive SST over the equatorial central-eastern Pacific during the OND season (Figure 1-A(a)). After 14 months, positive SST anomalies are observed over the equatorial central-eastern Pacific, indicating that the role of the western loading of SIOD-related SST anomalies on an El Niño event in the subsequent winter (Figure 1-A(b)). Furthermore, positive SST anomalies are shown over the equatorial central Pacific for the 14-months' partial lag regressions with respect to the eastern loading of the SIOD (Figure 1-A(c) and 1-A(d)).

In more detail, the amplitude of El Niño for the partial regression of the western loading of the SIOD SST is stronger to some extent than that of the eastern loading. The partial regression of low-level circulation anomalies support this notion that the partial regressed westerly anomalies over the equatorial western Pacific during the FMA(+1) season with respect to the western loading of the SIOD are stronger in amplitude compared to that of the partial regressed fields on the eastern loading of the SIOD which resembles the Ningaloo Niña (Figure 1-B).

This indicates that, while both the western and eastern loading of the SIOD plays a role to lead El Niño events in the subsequent year, the western loading of SIOD-related SST would induce the stronger influence to the equatorial Pacific. This implies that, only with cold SST anomalies over the southeastern Indian Ocean during the Ningaloo Niña, the remote impact of the southeastern Indian Ocean SST on the equatorial Pacific would be relatively weak.

The aforementioned findings are described in the revised manuscript as follows, and the Figure 1-A is added as Supp. Fig. 8.

Line 121-123: “The stronger impact of the SIOD on ENSO events in the subsequent year compared to that of Ningaloo Niño/Niña is because both the western and eastern loading of the SIOD has an additive effect on the ENSO (Fig. S8).”

Figure 1-A. Lagged partial regressions of (a) SST ($^{\circ}\text{C}$) anomalies during the OND and (b) subsequent year's SST ($^{\circ}\text{C}$) anomalies during the DJF season with respect to the western loading of the SIOD after excluding the impact of the eastern loading of the SIOD from 1998–2019. (c,d) As in (a,b), but for using the eastern loading of the SIOD after excluding the impact of the western loading of the SIOD. We multiplied -1 to the regressed fields for the eastern loading of the SIOD to match the sign of Ningaloo Niña event.

Figure 1-B. (a) Lagged partial regression of 925 hPa wind (vector, $\text{m}\cdot\text{s}^{-1}$) and streamfunction (contours at intervals of $0.3\cdot 10^{-5} \text{ m}^2\cdot\text{s}^{-1}$) anomalies during the FMA(+1) season with respect to the western loading of the SIOD after excluding the impact of the eastern loading of the SIOD from 1998–2019. (b,c) As in (a), but for using the eastern loading of the SIOD after excluding the impact of the western loading of the SIOD and the Ningaloo Niña, respectively. The shading and vector denote the region where the statistical significance is above the 95% confidence level. We multiplied -1 to the regressed fields for (b) and (c).

2. Based on the above, the authors should retreat from claiming the SIOD. Rather, they should focus on the importance of the SST anomalies in the Ningaloo Niño/Niña region. Even so, the punch line of the present work will be still alive as it is. Then, the title may be something like “Southeastern Indian Ocean SST anomalies as a trigger for Central Pacific El Niño since the 2000s” or simply “Ningaloo Niña as a trigger for Central Pacific El Niño since the 2000s”.

[Reply] Thank you for the reviewer’s valuable comment. Even though there is a similarity in the spatial distribution, we still think that the SIOD is a physical mode, whose mechanism is different from the Ningaloo Niño/Niña. To address this point, we firstly applied self-organizing maps (SOM) analysis (Tozuka et al., 2008) to the OND SST anomalies over the southern Indian Ocean (60° – 120° E, 30° S– 0°) for the period of 1976-2019 (total of 44 years).

The SOM is an unsupervised artificial neural network and shown to be an effective method for feature extraction from high dimensional data through classification into low dimensional data. As a reviewer pointed out, we are away from a danger of discussing a statistical artifact as in the EOF analysis, since artificial nodes that appear in a SOM array is never occupied (Liu et al., 2006). To reduce the input dimension and to focus on the large-scale SST variability, the resolution of the SST anomalies is coarsed to 10° (longitude) \times 5° (latitude) as in Tozuka et al. (2008). Therefore, each input vector has a 7×6 array, which consists of 39 grid points with defined values and three grid points with undefined values for Australia. The output vectors into which the input vectors are classified are composed of 2×3 types (total of 6 nodes) to have enough years to construct composite diagrams. As a result, the OND SST anomalies over the southern Indian Ocean are classified into six types based on the Euclidian distance. Each node in the SOM array is associated with a reference vector with dimensions equal to that of the input vector, i.e., 42 and the “rectangular” lattice structure is used as the shape of the network. Note that the output vectors are initialized with random values that are evenly distributed in the area of corresponding data vector components.

The years and number of events for each of six types are listed in Table 1-A, and the composited SST patterns for the corresponding types are shown in Figure 1-C. Type 1, and Type 6 exhibits a basin-wide warming and cooling SST anomaly pattern, while Type 3, and Type 5 represent east-west oriented SST dipole patterns associated with negative, and positive SIOD events, respectively. Type 2, and Type 4 are similar to that of the IOSD, and IOD mode, respectively. These results suggest that the SOM successfully captures the east-west-oriented SST dipole pattern associated with the SIOD mode during the OND season. This supports our notion that the SIOD is a physical mode over the southern Indian Ocean. This point is mentioned in Line 69-71 of the revised manuscript.

Table 1-A. Years and number of events for each of six types in the SOM.

Type	Years	Number of events
1	1979, 1990, 2013, 2014, 2015	5
2	1981, 1982, 1983, 1988, 1999, 2000, 2001, 2002, 2009, 2011, 2012	11
3	1984, 1989, 1996, 1998, 2010	5
4	1977, 1991, 1994, 1997, 2006, 2019	6
5	1976, 1978, 1980, 1986, 1987, 2003, 2008, 2017, 2018	9
6	1985, 1992, 1993, 1995, 2004, 2005, 2007, 2016	8

Figure 1-C. Composite of observed SST (°C) anomalies during the OND season for Types 1-6 from 1976–2019 as revealed by the SOM. A number in the upper left corner on each panel denotes a type index.

To examine the detailed physical mechanisms of the SIOD, we first defined the positive SIOD events as the years of Type 5. The selected years for the positive SIOD events are 1976, 1978, 1980, 1986, 1987, 2003, 2008, 2017, and 2018. And we adopted the definition of Kataoka et al. (2014) to select the year of Ningaloo Niña events. The selected years are 1986, 1990, 1991, 1992, 2003, 2004, and 2005. During 1980-2019, only two cases out of seven SIOD events (28.5%) is overlapped with the years of Ningaloo Niña events.

In addition, the composited SST patterns during the positive SIOD events showed a different time evolution from those during Ningaloo Niña events (Figure 1-D). That is, even though the SIOD-related cold SST anomaly over the southeastern Indian Ocean is quite similar to that of the Ningaloo Niña during the peak season (i.e., the OND season), the spatial distributions of the SST anomalies during the developing seasons are systematically different. During the preceding AMJ and JAS season, the negative SST anomalies over the eastern Indian Ocean are meridionally elongated from the equator to 45°S for the SIOD events. On the other hand, the negative SST anomalies are confined over the west coast of Australia during the Ningaloo Niña events, instead, it is zonally extended to the southcentral Indian Ocean.

To emphasize the different dynamical and thermodynamical processes for the positive SIOD events from that of the Ningaloo Niña events, we performed a heat budget analysis for the mixed-layer temperature anomalies as follows (Morioka et al., 2010).

$$\delta \left(\frac{\partial T_{mix}}{\partial t} \right) = \delta \left(\frac{Q}{\rho c_p H} \right) + \delta(\text{oceanic advection terms})$$

Here, ρ is the density, c_p is the specific heat of sea water, Q is the net surface heat flux, and $\delta()$ corresponds to the deviation from the monthly climatology. The mixed-layer depth is defined as the depth at which temperature is 0.8°C lower than the SST. According to the equation, the change rate of the anomalous mixed-layer temperature (T_{mix}) is led by the net surface heat flux (NSHF) anomaly and the anomalous oceanic temperature advections.

Figure 1-E shows the time evolution of the summed oceanic temperature advection terms, and NSHF anomaly and its decomposed terms (i.e., shortwave radiation, longwave radiation, latent heat flux, and sensible heat flux) during the positive SIOD events. For the western pole (65°–85° E, 25° S–10° S) of positive SIOD, the warming tendency of the anomalous mixed-layer temperature grows remarkably during late spring owing to the NSHF anomaly (Figure 1-E(a)). In particular, the latent heat flux anomaly plays a dominant role (Figure 1-E(b)). Also, the cooling tendency of the anomalous mixed-layer temperature over the eastern pole (90° – 120° E, 30° S – 5° S), which is strongest in boreal fall, is mostly determined by the latent heat

flux anomaly (Figure 1-E(c) and 1-E(d)).

During the Ningaloo Niña events, the weak cooling tendency of the anomalous mixed-layer temperature starts to grow from boreal summer due to the oceanic advection over the region of southwestern Indian Ocean (i.e., region for the western loading of SIOD-associated SST anomalies) (Figure 1-F(a)). During boreal fall, contribution of the NSHF anomaly becomes prominent, which are mostly attributed by the shortwave radiation anomaly (Figure 1-F(b)). The relatively important role of the shortwave radiation anomaly is also clear for the Ningaloo Niña region (i.e., defined same as the eastern loading of the SIOD-related anomalies) (Figure 1-F(c), and 1-F(d)), which is consistent with previous study (Kataoka et al., 2017). These results demonstrate that the physical mechanisms of the SIOD events are distinct from that of the Ningaloo Niña events.

The stronger contribution of the latent heat flux anomaly to the development of the positive SIOD-related cold SST anomalies is consistent with the robust dipole structure of the meridional wind anomalies associated with the large-scale anticyclonic circulation over the southern Indian Ocean (Figure 1-G); the southerly anomalies over the southeastern Indian Ocean enhance the wind speed to increase the evaporation, therefore, latent heat flux into the atmosphere is intensified to reduce the SST. On the other hand, the northerly anomalies are prominent over the southcentral Indian Ocean from the AMJ season to the OND season to increase the SST. While the dipole pattern of the latent heat flux anomalies is clear from the AMJ season to the OND season, however, the negative shortwave radiation anomalies are shown only at the OND season.

Even though the anticyclonic flow anomalies are also observed during the Ningaloo Niña events (Figure 1-H), however, its amplitude is systematically weaker than that during the SIOD events. In addition, during the AMJ season, cyclonic wind anomalies are shown over the eastern Indian Ocean, indicating that the dynamical coupling between the anticyclonic flow and the SST anomalies during the Ningaloo Niña events is not strong as those during the SIOD events. Consistent with the weaker meridional wind anomalies, the dipole structure of the latent heat flux anomalies over the western and eastern Indian Ocean is not shown for the Ningaloo Niña events. Instead, the negative shortwave radiation anomalies are evident over the eastern Indian Ocean from the JAS season, which confirms a heat budget analysis.

The aforementioned differences shown in the SOM, heat budget, and composite analysis indicate that the physical processes responsible for the positive SIOD events are distinct from the Ningaloo Niña events. Thus, it can be considered that the SIOD is a new climate mode

independent to the Ningaloo Niño/Niña to some extent.

The aforementioned discussions are added in the revised manuscript as follows. In addition, the Figure 1-D, Figure 1-E, and Figure 1-F are added as Supp. Figs. S4, S5, and S6.

Line 76-78: “In addition, even though there is a similarity in the spatial distribution between the positive SIOD and Ningaloo Niña events (Fig. S4), the physical mechanisms of the two are distinct (Figs. S5 and S6).”

Figure 1-D. Composite of observed SST ($^{\circ}\text{C}$) anomalies during the (a) AMJ, (b) JAS, and (c) OND season for positive SIOD events from 1980–2019. (d-f) are identical with (a-c) except for the Ningaloo Niña events.

Figure 1-E. (a) Heat budget anomaly of the mixed-layer temperature (left, °C/month), and (b) surface heat flux (right, °C/month) anomalies in the western pole (65°–85° E, 25° S–10° S) for positive SIOD events (Type 5) from 1980–2019. In the left panel, black line shows the change rate of the anomalous mixed-layer temperature, red line represents the net surface heat flux anomaly, and blue line indicates the summed oceanic temperature advection terms. In the right panel, red, purple, orange, blue, and green lines correspond to the net surface heat flux (Qnet), shortwave radiation (SW), longwave radiation (LW), latent heat flux (LH), and sensible heat flux (SH) anomalies, respectively. (c,d) As in (a,b), but for the eastern pole (90°–120° E, 30° S–5° S). Note that a positive sign of the heat flux terms denotes an additional heat flux into the ocean.

Figure 1-F. Same as Figure 1-E but for Ningaloo Niña events.

Figure 1-G. Composite maps of observed OLR ($\text{W}\cdot\text{m}^{-2}$; shading), 925 hPa wind ($\text{m}\cdot\text{s}^{-1}$; vectors), and streamfunction (contours at intervals of $0.3\cdot 10^{-5} \text{ m}^2\cdot\text{s}^{-1}$) anomalies during the (a) AMJ, (b) JAS, and (c) OND season for positive SIOD events from 1980–2019. (d-f and g-i) are identical with (a-c) except for latent heat flux and shortwave radiation.

Figure 1-H. Same as Figure 1-D but for Ningaloo Niña events.

3. Finally, I note that Ningaloo Nina events are more frequent in recent decades as partly seen in Table 1 of Tozuka et al. (2014). Also, Feng et al. (2021, *J. Marine Systems*, <https://doi.org/10.1016/j.jmarsys.2020.103473>) have recently discussed multi-year marine cold-spells off the west coast of Australia. Those seem to support the punchline of the authors.

[Reply] Thank you very much for proving related the references. Those references are cited in the revised manuscript as below:

Line 58-60: “On the other hand, SST variability over the southeastern IO has increased considerably since the late 1990s, which can influence tropical Pacific variability through interbasin coupling mechanisms(Tozuka et al., 2014, Feng et al., 2015; Zhang and Han, 2018; Feng et al., 2021).”

Minor comment

1. Line 119: cold SST anomalies -> cold/warm SST anomalies

[Reply] Corrected.

References

- Feng, M., N. Caputi, A. Chandrapavan, M. Chen, A. Hart, and M. Kangas, 2021: Multi-year marine cold-spells off the west coast of Australia and effects on fisheries. *J. Marine Systems*, 214, 103473.
- Feng, M., and Coauthors, 2015: Decadal increase in Ningaloo Niño since the late 1990s. *Geophys. Res. Lett.*, 42, 104-112.
- Kataoka, T., T. Tozuka, and T. Yamagata, 2017: Generation and decay mechanisms of Ningaloo Niño/Niña. *J. Geophys. Res. Ocean.*, 122, 8913-8932.
- Kataoka, T., T. Tozuka, S. Behera, and T. Yamagata, 2014: On the Ningaloo Niño/Niña. *Clim. Dynam.*, 43, 1463-1482.
- Liu, Y., R. H. Weisberg, and C. N. Mooers, 2006: Performance evaluation of the self-organizing map for feature extraction. *J. Geophys. Res. Ocean.*, 111.
- Morioka, Y., T. Tozuka, and T. Yamagata, 2010: Climate variability in the southern Indian Ocean as revealed by self-organizing maps. *Clim. Dynam.*, 35, 1059-1072.
- Tozuka, T., T. Kataoka, and T. Yamagata, 2014: Locally and remotely forced atmospheric circulation anomalies of Ningaloo Niño/Niña. *Clim. Dynam.*, 43, 2197-2205.
- Tozuka, T., J.-J. Luo, S. Masson, and T. Yamagata, 2008: Tropical Indian Ocean variability revealed by self-organizing maps. *Clim. Dynam.*, 31, 333-343.
- Zhang, L., and W. Han, 2018: Impact of Ningaloo Niño on tropical Pacific and an interbasin coupling mechanism. *Geophys. Res. Lett.*, 45, 11,300-311,309.

REVIEWER COMMENTS:

Reviewer #3:

The authors have mostly addressed previous concerns. While the response letter provides reasoning for the choices made regarding the analysis period (i.e., after the mid-1970s climate shift in the Indo-Pacific) and a sentence to that effect has been added in L262-264, the broader issue has not been addressed: ie. the authors are encouraged to embed their results into a broader discussion of multi-decadal variability in the Indo-Pacific and implications for ENSO characteristics and predictive skill. This would greatly strengthen the study and raise its appeal to a broader readership. As it stands, the value is more narrowly focused on a readership with more discipline-specific interest in seasonal forecasting, as well as ENSO predictions and evolution.

[Reply] We appreciate the reviewer's constructive comments. To reflect the reviewer's suggestions, we added the following discussions in the revised manuscript:

Role of the SIOD on Pacific decadal variability

Line 243-245: "..., whereas the SIOD promotes the persistence of the ENSO for two consecutive years. Therefore, the SIOD possibly contributes to the long-lived SST anomalies over the central Pacific(Kug et al., 2009), which are responsible for the Pacific decadal variability(Power et al., 1999; Meehl et al., 2013)."

Role of the SIOD on ENSO characteristics

Line 252-254: "As the SIOD modulates both the spatial and temporal characteristics of the ENSO, advances in research towards a better understanding of the role of the SIOD may strengthen its potential as a unique and reliable predictor for CP-type El Niño events in recent decades."

Role of the SIOD on predictive skill of the global climate variabilities

Line 254-257: "Therefore, in addition to the countries adjacent to the equatorial Pacific, the main findings of this study have the potential to enhance the forecasting ability for global climate variabilities led by various types of El Niño-induced atmospheric teleconnections(Yeh et al., 2009)."

The supplementary material provides an extensive set of additional information for interested readers. Given that Nature Communications allows for a larger number of display items as part of the main text, one consideration for the authors would be to add some of that material to the main part of the manuscript. This would improve the readability of the manuscript considerably, rather than referring extensively to the supplement. The writing has improved to the previous

version; however, the manuscript throughout would still benefit from further input from a native English speaker. Some specific examples for rewording are given below; however, this is not an exhaustive list.

[Reply] Thank you for the reviewer’s comment. We agree that some of materials in the supplementary figures are worthwhile to be moved to the main figure for the readers to understand our main point. To clearly demonstrate recent changes in the dominant oceanic variability over the southern Indian Ocean and its coupling to the subsequent year’s ENSO events, we have added Supplementary Fig. 1 in the previous round of the review (also Figure 2-A in the rebuttal letter) as the main Figure 1. And, we conducted a substantial copy-editing by a native English speaker throughout the manuscript.

Figure 2-A. The emergence of a new climate mode over the Southern Indian Ocean after the 2000s. **a**, 15-yr moving correlation coefficients between the IOD index during the SON season, the IOBM index during the DJF season, and the subsequent year’s Niño3.4 index during the DJF season (blue line: IOD vs. Niño3.4; red line: IOBM vs. Niño3.4) from 1976-2019. The x -axis indicates the middle year in the 15-yr moving window (e.g., 1999 indicates the correlation coefficient from 1992–2006). The dots denote correlation coefficients that are statistically significant at the 95% confidence level. **b**, Second EOF of the OND SST anomalies over the IO for 1976-1997. **c**, As in **b**, but for 1998–2019. The units for the EOF are non-dimensional.

Specific comments

1. L50: Reword to “... and hence ENSO phase transitions becoming less regular...”

[Reply] Corrected

2. L52: Reword to "... and it is therefore..."

[Reply] Corrected

3. L60: Reword to "... has markedly increased in frequency, which..."

[Reply] Corrected

4. L70: Reword to "... which appears as..."

[Reply] Corrected

5. L78: That seems contradictory that SIOD peak amplitude is delayed and ahead in time? Please clarify.

[Reply] Sorry for the confusion about the sentence. We tried to explain that the SIOD peak season is delayed than that of the IOD peak season, and ahead than that of the IOSD peak season. To avoid the confusion, we deleted the corresponding sentence and added the following sentence in the revised manuscript:

Line 74-76: "Note that the SIOD is independent of the IOSD and the IOD in terms of the temporal lead-lag correlation (Fig. S2) and the season of peak intensity (Fig. S3)."

6. L117: Reword to "... off Australia..."

[Reply] Corrected

7. L247: Should be "avenue" rather than "revenue"?

[Reply] Corrected

8. Fig 1: Black boxes in a-d are hard to see/distinguish from other map features. Consider using a distinct color to better highlight the boxes.

[Reply] We have replaced the black boxes in Figure 1 with the purple boxes in Figure 2 in the revised manuscript to better highlight the boxes as you pointed out.

References

- Kug, J.-S., F.-F. Jin, and S.-I. An, 2009: Two types of El Niño events: cold tongue El Niño and warm pool El Niño. *J. Clim.*, 22, 1499-1515.
- Meehl, G. A., A. Hu, J. M. Arblaster, J. Fasullo, and K. E. Trenberth, 2013: Externally forced and internally generated decadal climate variability associated with the Interdecadal Pacific Oscillation. *J. Clim.*, 26, 7298-7310.
- Power, S., T. Casey, C. Folland, A. Colman, and V. Mehta, 1999: Inter-decadal modulation of the impact of ENSO on Australia. *Clim. Dynam.*, 15, 319-324.
- Yeh, S. W., J. S. Kug, B. Dewitte, M. H. Kwon, B. P. Kirtman, and F. F. Jin, 2009: El Niño in a changing climate. *Nature*, 461, 511-514.

REVIEWERS' COMMENTS

Reviewer #2 (Remarks to the Author):

Review of "Southern Indian Ocean Dipole as a trigger for Central Pacific El Niño since the 2000s" by Hyun-Su Jo et al.

I have read through the revised version of this interesting article submitted to Nature Communications. I acknowledge that the authors have sincerely responded to my critical comments in the second-round review.

I'm still not entirely convinced by the author's claim of a new climate mode of SIOD as a physical mode triggering the Central Pacific El Niño since the 2000s. For example, Fig. 2d for SIOD and Fig. S7 b for NN look almost exactly the same to me, in contrast to the authors' claim that the NN influence is negligible (l. 120-l. 121). Actually, the authors admit the SIOD has an additive effect on the ENSO (l. 123).

However, judgment should be left to the discerning readers. I have no objection to the publication of this paper if the authors address the following minor points.

- l. 77: Ningaloo Niña suddenly appeared here without any reference.
- l. 132: Fig. 2f is for 1998-2019, not after the 2000s.
- l. 144: Kelvin wave should be stationary Kelvin wave.
- l. 213: The original article by Ashok et al. (2007) (Ref. 11) should be cited as well.

REVIEWER COMMENTS:

Reviewer #2:

I have read through the revised version of this interesting article submitted to Nature Communications. I acknowledge that the authors have sincerely responded to my critical comments in the second-round review.

I'm still not entirely convinced by the author's claim of a new climate mode of SIOD as a physical mode triggering the Central Pacific El Niño since the 2000s. For example, Fig. 2d for SIOD and Fig. S7 b for NN look almost exactly the same to me, in contrast to the authors' claim that the NN influence is negligible (l. 120-l. 121). Actually, the authors admit the SIOD has an additive effect on the ENSO (l. 123).

However, judgment should be left to the discerning readers. I have no objection to the publication of this paper if the authors address the following minor points.

[Reply] Thank you for providing additional comments. We have revised all comments raised by the reviewer.

Minor comments

l. 77: Ningaloo Niña suddenly appeared here without any reference.

[Reply] References related to Ningaloo Niña are added in the revised manuscript.

l. 132: Fig. 2f is for 1998-2019, not after the 2000s.

[Reply] Corrected

l. 144: Kelvin wave should be stationary Kelvin wave.

[Reply] Corrected

l. 213: The original article by Ashok et al. (2007) (Ref. 11) should be cited as well.

[Reply] The reference is cited in the revised manuscript.